# Lattice-contraction triggered synchronous electrochromic actuator

Kerui Li[1], Yuanlong Shao[2], Hongping Yan[3], Zhi Lu[4], Kent J. Griffith [5], Jinhui Yan[6], Gang Wang [7], Hongwei Fan[1], Jingyu Lu[5], Wei Huang [8], Bin Bao[2], Xuelong Liu[1], Chengyi Hou[1], Qinghong Zhang[1], Yaogang Li[1], Junsheng Yu[8] & Hongzhi Wang[1]

Materials with synchronous capabilities of color change and actuation have prospects for application in biomimetic dual-stealth camouflage and artificial intelligence. However, color/shape dual-responsive devices involve stimuli that are difficult to control such as gas, light or magnetism, and the devices show poor coordination. Here, a flexible composite film with electrochromic/actuating (238° bending angle) dual-responsive phenomena, excellent reversibility, high synchronization, and fast response speed (< 5 s) utilizes a single active component, $W_{18}O_{49}$ nanowires. From in situ synchrotron X-ray diffraction, first principles calculations/numerical simulations, and a series of control experiments, the actuating mechanism for macroscopic deformation is elucidated as pseudocapacitance-based reversible lattice contraction/recovery of $W_{18}O_{49}$ nanowires (i.e. nanostructure change at the atomic level) during lithium ion intercalation/de-intercalation. In addition, we demonstrate the $W_{18}O_{49}$ nanowires in a solid-state ionic polymer-metal composite actuator that operates stably in air with a significant pseudocapacitive actuation.

[1] State Key Laboratory for Modification of Chemical Fibers and Polymer Materials, College of Materials Science and Engineering, Donghua University, Shanghai 201620, People's Republic of China. [2] Cambridge Graphene Center, Department of Engineering, University of Cambridge, Cambridge CB3 0FA, UK. [3] Stanford Synchrotron Radiation Light Source, SLAC National Accelerator Laboratory, Menlo Park 94025 CA, USA. [4] Department of Materials Science and Engineering, Northwestern University, Illinois 60208, USA. [5] Department of Chemistry, University of Cambridge, Cambridge CB2 1EW, UK. [6] Department of Mechanical Engineering, Northwestern University, Illinois 60208, USA. [7] School of Chemical and Biomolecular Engineering, School of Chemistry and Biochemistry, School of Materials Science and Engineering, Georgia Institute of Technology, Atlanta 30332 GA, USA. [8] State Key Laboratory of Electronic Thin Films and Integrated Devices, School of Optoelectronic Information, University of Electronic Science and Technology of China, Chengdu 610054, People's Republic of China. These authors contributed equally: Kerui Li, Yuanlong Shao. Correspondence and requests for materials should be addressed to G.W. (email: gang.wang@northwestern.edu) or to H.W. (email: wanghz@dhu.edu.cn)

Chameleon camouflage, a biomimetic technology derived from the reversible shift of skin colors of chameleons through controlling the lattice of guanine nanocrystals[1], has been extensively studied to improve current camouflage methods originating from pattern painting dating back to the 1940s[2–4]. Although chameleons have been widely accepted as "masters of disguise", there are still some other natural color-changers including octopus vulgaris, cuttlefish, and the Andean rainfrog, which show more sophisticated camouflage abilities[5–7]. They can not only adapt rapidly to the surrounding colors just like chameleons, but also reversibly change their texture or posture to fit the environments, which results in a greater chance of survival.

Recently, inspired by animals with dual-stealth capability, several research groups have been dedicated to developing smart hybrid materials driven by different stimuli (such as gas, light, magnetism, and so on), to realize the biomimetically dual-responsive technology[8–11]. For instance, Kwak and colleagues reported a fluorescent actuator based on a bilayer stack structure in response to ethanol vapor[9]. Naumov and colleagues reported a light- and humidity-induced actuator consisting of an acid-ochromic fluorophore guest and agarose which also had acid-induced fluorescence changes[10]. Kwon and colleagues fabricated a magnetochromatic microactuator through self-assembling superparamagnetic colloidal nanocrystal clusters in a photocurable polymer resin[11]. Despite these achievements, there still exist some critical shortcomings: (i) the dual-responsive performances of these materials were stimulated by gas, light or magnetism, which are affected by environmental conditions and are weakly controllable; (ii) the color changes were not striking enough to be observed by the naked eye within all visual angles, which lead to moderate detectability.

Compared with the stimuli described above, electricity is an easily and efficiently controllable input factor. Among many electro-responsive materials, there is some scope to realize both electrochromic (EC) and electrochemical actuating functions based on a single device because of their similar reaction and operation conditions[12–15]. Recently, Hosson and colleagues developed this type of dual-responsive composite film by electrodepositing an EC material (polyaniline) on an electrochemical actuator (nanoporous gold film)[16]. This dual-responsive composite film can not only be efficiently controlled by an input electrical signal, but also exhibit many other advantages such as low driving voltage, high reversibility, and fast response rate. However, the composite films were restricted by the mutual influence of the colors between two functional materials, poor interfacial stability as well as small deformation of nanoporous gold films. Due to huge mechanistic difference between the nonfaradaic actuation and faradaic EC effect[12,14,17,18], the composite structure of two functional materials is only one approach to realize the electro-induced dual-responsive effect and difficult to be improved to solve the problems described above at present. Therefore, although the electrical stimulation has shown high efficiency and controllability, a strategy to combine high-performance electrochromism with electrochemical actuation *via* a unique material or structure becomes the key for this biomimetic technology.

Here, the reversible pseudocapacitive EC deformation of $W_{18}O_{49}$ nanowires ($W_{18}O_{49}$NWs) induced by lattice contraction/recovery is found during Li ion ($Li^+$) intercalation/de-intercalation processes. The actuating mechanism is elaborated and further verified through well-designed control experiments, first principles calculation, in-/ex-situ microstructure characterization, and numerical simulations using Isogeometric Analysis. Therefore, a highly flexible dual-responsive film based on Ag nanowires (AgNWs)/$W_{18}O_{49}$NWs bilayer networks is fabricated and demonstrates great synchronous EC and actuating performances. In addition, a symmetric ionic polymer metal composite (IPMC) actuator is built based on pseudocapacitive actuation of $W_{18}O_{49}$NWs to extend its application range in air and some other complex environments, which introduces a new approach for building IPMC and achieving enhanced properties.

## Results

**Structural characterization of as-prepared composite films.** The $W_{18}O_{49}$NWs display diameters centered around 20 nm (Fig. 1a and Supplementary Fig. 1a) and lengths of ca. 12 μm (Supplementary Figs. 1b, c), which results in an aspect ratio of around 600. This high aspect ratio is beneficial to the improvement of their intrinsic flexibility and formation of highly connected networks. The X-ray diffraction peaks (Supplementary Fig. 1d) can be clearly indexed as monoclinic $W_{18}O_{49}$ (*P2/m*, JCPDS No. 84-1516, ICSD 202488). The narrow (010) and (020) peaks with relatively high intensity strongly suggest that crystals grow along the direction of [010]. In the high-resolution transmission electron microscopy (HRTEM) image (Fig. 1a), the lattice fringes with a wide spacing of 0.378 nm and regularly layered lattice structure along the direction of [010] can be also clearly observed. The wide lattice spacing and high aspect ratio can provide access to more active sites through favorable solid-state-ionic diffusion[19].

With these $W_{18}O_{49}$NWs as EC materials, a highly flexible dual-responsive film is prepared using a spray-coating method (Illustrated in Supplementary Fig. 1e). In the active area, ultrathin poly(3,4-ethylenedioxythiophene)-poly(styrenesulfonate) (PEDOT:PSS) is coated onto the pure AgNW network to protect the AgNWs from electrochemical corrosion. $W_{18}O_{49}$NW networks stack on AgNW/PEDOT:PSS networks, which forms a bilayer nanowire network (BNN) on a polyethylene terephthalate (PET) substrate with a thickness of ca. 15 μm. Figure 1b, c exhibit the surface and cross-section morphologies of the as-prepared dual-responsive film. The thickness of the bilayer nanowire network is only ~195 nm, which is much thinner than almost all of the EC active layers reported in previous literature[20–22]. This ultra-thin active layer could efficiently improve the bendability and foldability according to following equation[23],

$$R_{min} = \frac{b}{2}\left(\frac{1}{\Delta L} - 1\right), \qquad (1)$$

where $R_{min}$ is the radius of curvature under maximum bending, $b$ is the diameter or thickness, $\Delta L$ is the elongation at break ($0 < \Delta L < 1$). To further improve the interfacial and structural stability, mussel-inspired alginic acid/poly(dopamine) (Aa-PDA) complex was utilized as a binder. As a result, the sheet resistance of the conductive layer remained almost constant during 1000 bending cycles (bending radius of 2.5 mm) and changed only slightly even after 100 folding cycles and peeling cycles using 3M Scotch tape, respectively (Supplementary Figs. 2a, b). Moreover, the EC contrast only slightly decreased after 100 folding cycles (Supplementary Fig. 2c). Therefore, BNNs can remain highly connective without large displacement during complex mechanical deformation processes, which ensures stable actuating performance. Further, as shown in Supplementary Fig. 2d, the dual-responsive film demonstrates an optical transmittance of 71.2% at 633 nm, while the conductive layer shows a very low sheet resistance of ca. 9.0 ohm sq$^{-1}$. These initial values can fully meet the requirement of EC electrodes for optical transmittance and conductivity.

**The electrochromic and actuating performances of the composite films.** The dual-responsive film was used as a working

electrode for the EC/actuating dual-responsive measurements with Pt wire (0.5 mm diameter) as the counter electrode, Ag/AgCl (3.0 M KCl) as the reference electrode, and 1 M lithium perchlorate (LiClO₄)/propylene carbonate (PC) solution as the electrolyte (Fig. 1d and Supplementary Fig. 3). As shown in Fig. 1e and Supplementary Movie 1, the dual-responsive film shows great synchronism and excellent EC/actuating performances. The variation in the optical transmittance at 633 nm between the original state and −0.9 V reached 29.9%, while it exhibited a large variation of 46.1% at −1.8 V (Fig. 1f). The dual-responsive film was completely bleached when the low positive voltage of +0.6 V was applied, indicating its great optical reversibility. The optical switching speed between states is generally defined as the time for a 90% change in the entire transmittance modulation. As observed in both current and transmittance response curves (Fig. 1g, h), the dual-responsive film presents dramatically fast switching speed. The colored and bleached times are calculated to be 4.1 and 2.9 s, respectively, which are much shorter than most results of WO₃-based EC electrodes[24–28]. Coloration efficiency (CE), which represents the change in optical density (OD) per unit charge density ($Q/A$) during switching, is a crucial criterion for the practical application of EC devices. According to the formulae, $CE = \frac{\Delta OD}{Q/A}$, the CE of dual-responsive film was calculated to be 119.2 cm² C⁻¹ (Fig. 1i), which is much higher than previously reported WO₃-based values tested under

same conditions and is comparable to the results tested even in aqueous electrolytes despite that aqueous systems are generally considered to be advantageous for ion diffusion in the electrolyte[23].

Besides excellent electrochromic properties, the dual-responsive film demonstrates great actuating performance. As shown in Fig. 1e, the large deformation of BNNs resulted in the obvious bending motion. Thus, we use the bending angle to evaluate the deformation effect of the dual-responsive film. Remarkably, the bending angles of the film quickly increased and reached a maximum at 238° within only 5 s when a constant voltage of −0.9 V was applied (Fig. 1g). To the best of our knowledge, this bending angle is higher than almost all the electrochemical actuators[17,18,29,30] and even comparable to best reported results of actuators stimulated by other factors such as illumination, heat, moisture, chemicals and so on[31–34]. The maximum curvature and strain were also calculated to be 1.22 cm⁻¹ and 1.81%, respectively, according to Supplementary Notes 1 and 2[35]. The maximum actuating curvature, strain, and bending angle are higher than those of most reported conducting polymer-based actuators with similar bilayer structure and reaction conditions (Supplementary Table 1). This high dual-responsive performance can be attributed to the highly interconnected three-dimensional micro-/nano-channels for highly efficient ion diffusion and penetration. In addition, highly

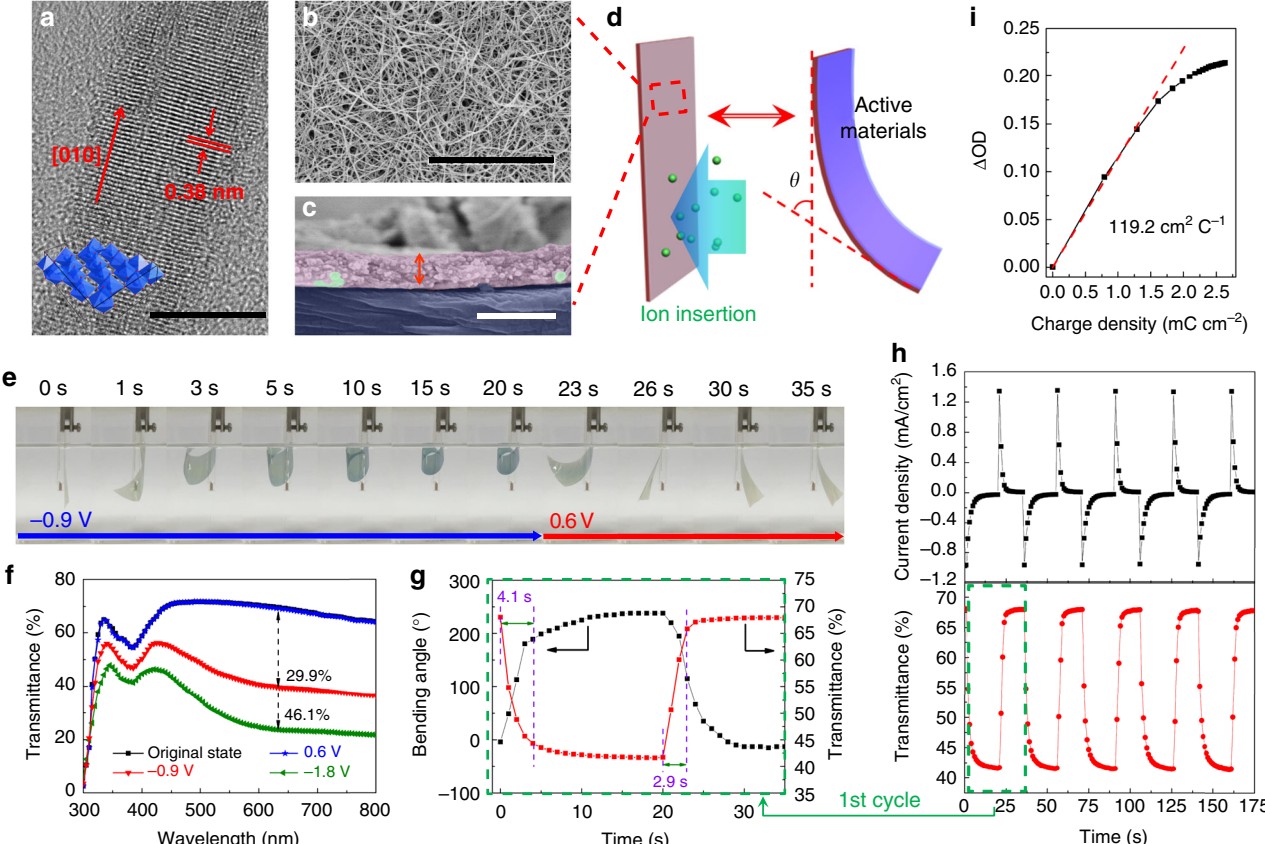

**Fig. 1** The dual-responsive performances of as-prepared composite films. **a** High-resolution transmission electron microscopy (HRTEM) image of a single W₁₈O₄₉ nanowire (W₁₈O₄₉NW) (Scale bar: 10 nm); Field emission scanning electron microscopy (FE-SEM) images of **b** surface (Scale bar: 3 μm) and **c** cross-section (Scale bar: 500 nm) of the dual-responsive film; **d** Schematic illustration of measurement criteria of electrode deformation angles during the electrochemical reaction process in 1 M LiClO₄/propylene carbonate (PC) electrolyte; **e** Digital photographs of synchronous electrochromic/actuating processes of the dual-responsive film. **f** Ultraviolet–visible (UV-vis) transmittance spectra of the dual-responsive film measured at the original state, −0.9, −1.8 and 0.6 V, respectively; **g** In situ enlarged transmittance response and deformation angle response between the colored and bleached states for the dual-responsive film measured at +0.6 and −0.9 V bias; **h** In situ current (up) and transmittance (bottom, at 633 nm) responses between the colored and bleached states. **i** Optical density (OD) as a function of charge density for the dual-responsive film

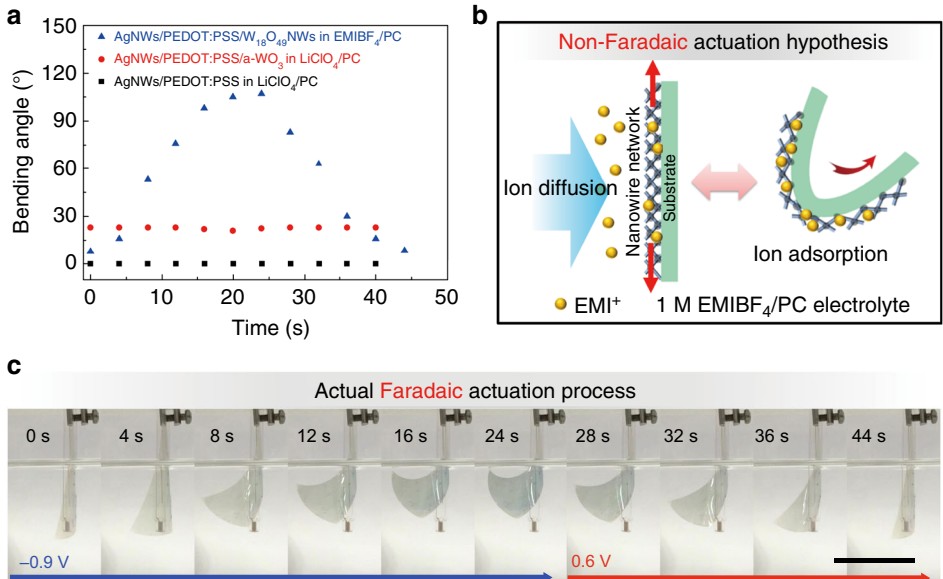

**Fig. 2** Three control experiments. **a** Actuating angles of different films as a function of time. **b** Theoretical schematic illustration of nonfaradaic reaction-based actuation for the dual-responsive films in 1 M 1-ethyl-3-methylimidazolium tetrafluoroborate (EMIBF$_4$)/ propylene carbonate (PC) electrolyte; **c** Digital photographs of electrochromic (EC) and deformation processes of the dual-responsive film (Scale bar: 3 cm) measured in 1 M EMIBF$_4$/PC electrolyte, respectively. (NWs is nanowires, PEDOT:PSS is poly(3,4-ethylenedioxythiophene)-poly(styrenesulfonate))

conductive AgNWs layer and connected networks ensure good electron transport between nanowires. Electrochemical stability was also characterized by chronoamperometry using square potentials between −0.9 and 0.6 V (Supplementary Fig. 4). Although the maximum bending angle decreased by ca. 24%, the peak current density and optical contrast of the composite film only decreased by less than 10% during 100 electrochemical cycles.

**Control experiments to determine the actuating materials.** According to previous research[21,23,25], the EC phenomenon of the dual-responsive films is mainly attributed to the reversible faradaic redox reactions of W$_{18}$O$_{49}$NWs, according to the following equation:

$$W_{18}O_{49} + xLi^+ + xe^- \leftrightarrow Li_xW_{18}O_{49}. \tag{2}$$

Recently, according to the difference of electrochemical processes, the actuating mechanisms of electrochemical actuators can be divided into two categories: nonfaradaic (electrostatic double-layer mechanism) and faradaic reactions (pseudocapacitive mechanism). For the dual-responsive films, there are many potential pseudocapacitive materials (W$_{18}$O$_{49}$NWs, PEDOT:PSS and AgNWs) and BNNs with a high specific surface area for the favorable electrostatic charge storage. Therefore, to confirm active materials and explain the possible mechanism for the outstanding actuation, we designed three parallel control experiments with different active components or electrolytes and the same applied electrochemical parameters.

First, the AgNWs/PEDOT:PSS composite film was tested in the 1 M LiClO$_4$/PC electrolyte. As shown in Fig. 2a and Supplementary Fig. 5, there is no deformation or displacement due to very low loading of PEDOT:PSS (one traditional actuating conducting polymer). Therefore, we can confirm the actuation is not caused by the AgNWs and PEDOT:PSS.

Second, the dual-responsive film was measured in 1 M 1-ethyl-3-methylimidazolium tetrafluoroborate (EMIBF$_4$)/PC electrolyte instead of LiClO$_4$/PC used above to determine whether the

actuation is caused by nonfaradaic reaction. According to the nonfaradaic actuating mechanism, if the negative voltage of −0.9 V is applied, the cations, EMI$^+$, with a large ionic radius will move into the porous channels of the electrodes and adsorb on the nanowires. As illustrated in Fig. 2b, theoretically, with the increase in the number of cations, the electrostatic repulsive interaction among the cations will expand the volume of BNNs, which leads to bending of the films to the right. However, in fact, under the voltage of −0.9 V, the dual-responsive films still bent to left, which is contrary to the theory based on nonfaradaic reaction (Fig. 2a, c). Moreover, Nobvious EC phenomenon was also observed, indicating the faradaic redox reaction happened between W$_{18}$O$_{49}$ and EMI$^+$. These phenomena prove that the electrochemical actuation should be attributed to the faradaic redox reactions of W$_{18}$O$_{49}$NWs instead of the electrostatic double-layer process.

Third, amorphous WO$_3$ (a-WO$_3$) was electrodeposited onto the AgNWs/PEDOT:PSS composite film and was also measured in LiClO$_4$/PC electrolyte to explore whether the change of crystal structure of W$_{18}$O$_{49}$NWs during faradaic reaction led to the actuation (Supplementary Fig. 6). Interestingly, as shown in Supplementary Fig. 6d, the AgNWs/PEDOT:PSS/a-WO$_3$ composite film slightly bent to the right, which is inconsistent with deformation of the previously tested dual-responsive film. This is probably due to expansion of a-WO$_3$ during the insertion of Li ions[36,37]. According to the analysis described above, we can confirm that the deformation of the layered crystal structure of W$_{18}$O$_{49}$NWs is the key factor for achieving the remarkable actuation during the pseudocapacitive process.

Supplementary Fig. 7 displays the cyclic voltammetry curves of different films. The dual-responsive film shows only one pair of redox peaks at the respective potentials of ca. −0.72 and −0.46 V under the scan rate of 5 mV s$^{-1}$. Such peaks are contributed from the redox reaction of W$_{18}$O$_{49}$ during Li$^+$ intercalation/de-intercalation, indicating that the main active ions are cations (i.e., Li$^+$ dominant reaction). In contrast, AgNWs/PEDOT:PSS composite films exhibited totally different CV curves with very broad redox peaks (Supplementary Fig. 7b), which are similar to

typical CV curves of PEDOT:PSS. The relatively small current densities of AgNWs/PEDOT:PSS composite films also prove the very low loading of PEDOT:PSS. For the dual-response films measured in 1 M EMIBF$_4$/PC electrolyte, there are relatively obvious redox peaks (Supplementary Fig. 7c) instead of the nearly-rectangular curves, which further supports the as-mentioned faradaic redox reaction between W$_{18}$O$_{49}$ and EMI$^+$. However, as shown in Supplementary Fig. 7d, under the same scan rate, the current densities of the dual-response films measured in 1 M EMIBF$_4$/PC electrolyte is lower than those measured in 1 M LiClO$_4$/PC electrolyte and the redox peaks are also shifted to higher potentials (ca. −0.9/0.015 V), indicating the poor solid-state diffusion of EMI$^+$ with its larger ionic radius.

**Density functional theory calculation for W$_{18}$O$_{49}$ nanowires**. To better understand the fast deformation mechanism of W$_{18}$O$_{49}$NWs at the atomic level, first principles calculations were used to simulate tNhe lithiation process and corresponding structural evolution. Crystalline W$_{18}$O$_{49}$ (WO$_{2.72}$) is known as an oxygen-deficient oxide with a most energetically favorable monoclinic structure, which can be regarded as WO$_3$ with some oxygen vacancies and distortion (Supplementary Fig. 8a). For the oxygen-deficient tungsten oxides, the content of oxygen-vacancies can be affected by many environmental factors. For example, under long-term storage or oxidative environments, the number of oxygen-vacancies will decrease[38,39]. When the tungsten oxides are annealed in the protective or reduced atmospheres, the number of oxygen-vacancies will increase[40,41]. Therefore, the actual content of oxygen-vacancies in W$_{18}$O$_{49}$NWs cannot be confirmed easily and precisely, which may influence the cell structure. Therefore, the cell structure of used W$_{18}$O$_{49}$NWs is perhaps slightly different from that of perfect monoclinic W$_{18}$O$_{49}$. Moreover, to simplify our calculation for monoclinic W$_{18}$O$_{49}$ with a complex cell structure, the density functional theory (DFT) calculation was based on the structure of mono-clinic WO$_3$. It is well known that the simplest structure of WO$_3$ is cubic, i.e. ReO$_3$ and is composed of corner-sharing regular octahedra with twelve-fold cavities in the polyhedra (Supplementary Fig. 8b). To explain the lithiation process of monoclinic WO$_3$ clearly, both the cubic and monoclinic structures were used as pristine cells (for the cubic cell, we build 2 × 2 × 2 supercell; for monoclinic W$_8$O$_{24}$, it could be regarded as 2 × 2 × 2 supercell of the cubic WO$_3$ as well). All the possible Li-vacancy configurations are enumerated with Li occupying the twelve-fold cavities in the polyhedral at various Li concentrations (Li$_x$WO$_3$, $x = 0$, 0.125, 0.25, 0.375, 0.5, 0.625, 0.75, 0.875, 1). The fully lithiated phase LiWO$_3$ is in the perovskites structure (ABO$_3$)[42–44].

Our calculation results indicate there is a significant volume (up to 5.4%) and interlayer (up to 2.0%) contraction with Li$^+$ insertion into the monoclinic WO$_3$ (Fig. 3a, b). However, the calculations in the cubic system display an opposite volume trend. The volume of cubic WO$_3$ increases with Li$^+$ insertion (Fig. 3c). By comparing the intermediate lithiated phases between the cubic and monoclinic systems, a large octahedral-tilting distortion is found in the monoclinic cell, while such structural evolution does not exist in the cubic cell. Meanwhile, the octahedra-tilting distortion helps to lower the energy and to convert the structure to a more stable phase with a denser packing (Supplementary Fig. 9). It is worth mentioning that this octahedra-tilting distortion has been well studied in perovskite structures (ABO$_3$) and commonly observed in a number of perovskites due to the size mismatch of the A and B cations[45,46]. Notably, our calculation also indicates only a small amount of inserted Li$^+$ can result in a large lattice contraction. For example, when WO$_3$ is just lithiated to Li$_{0.25}$WO$_3$, the cell reaches its maximum volume contraction. The obvious octahedra-tilting distortion triggered by a small amount of inserted Li$^+$ explains not only the large deformation but also the fast deformation rate during Li$^+$ insertion.

**Operando hard synchrotron X-ray diffraction of W$_{18}$O$_{49}$**. To further reveal the crystal structure transformation of W$_{18}$O$_{49}$NWs during the lithiation/delithiation processes and experimentally prove the first principles calculation results, operando electrochemical (de)lithiation of W$_{18}$O$_{49}$NWs and synchrotron diffraction measurements were performed with high energy X-rays (51.358 keV) for 1.5 cycles corresponding to lithiation–delithiation–lithiation. It is noted that due to the Li metal counter electrode used for the reference electrode in the in situ synchrotron X-ray diffraction (XRD) electrochemical cell, the potentials reported here are versus Li/Li$^+$, which is +3.25 V vs. Ag/AgCl. As shown in Fig. 3e, the galvanostatic electrochemistry was conducted from open-circuit voltage (3.5 V) to 2.0 V (i.e., −1.25 V vs. Ag/AgCl) on lithiation and to 3.8 V (i.e., 0.55 V vs. Ag/AgCl) on delithiation. Based on the favorable growth direction along the b-axis, the XRD pattern of W$_{18}$O$_{49}$ was dominated by crystallographic (010) and (020) reflections. The (h0l) reflections are severely size-broadened due to their nanoscale dimensions, which is consistent with the HRTEM results.

As shown in the enlarged XRD pattern from $2\theta = 3.5°$ to 3.8°, we clearly observed the reversible contraction/expansion of crystal lattice (010) spacing, i.e. the crystallographic b-axis (perpendicular to the layers in W$_{18}$O$_{49}$), during the electrochemical lithiation/delithiation processes. During the initial lithiation from 3.5 V to 2.23 V (i.e. −1.02 V vs. Ag/AgCl), it exhibits a gradual right-shift from 3.602° to 3.647°. Here, the mole ratio of intercalated Li$^+$ (relative to the WO$_{2.72}$, i.e., W$_{18}$O$_{49}$) can be calculated to be ~ 0.26 (Li$_{0.26}$WO$_{2.72}$), which is consistent with the DFT calculation result of the Li mole ratio (Li$_{0.25}$WO$_3$) required for maximum volume contraction. When the voltage continues to decrease to 2.0 V, the (0k0) reflections do not show obvious shifts and remain roughly constant, which is consistent with the previous DFT calculation. During the following delithiation and repeated lithiation process, the crystal lattice spacing of (010) expanded and contracted, demonstrating this process is reversible and repeatable.

**The structural characterization during the Li$^+$ insertion**. Under the same electrochemical reaction conditions, ex situ grazing incidence X-ray diffraction (GIXRD) and HRTEM were performed before and after the Li$^+$ insertion for the W$_{18}$O$_{49}$NWs in dual-responsive films. As shown in Supplementary Fig. 10, the (010) GIXRD peak shifts slightly to a higher-angle region, which indicates the contraction, rather than expansion, of the lattice spacing along the nanowire growth direction. Furthermore, the contraction was observed by the reduction in spacing of the (010) plane from 0.378 nm to 0.374 nm in HRTEM images after the intercalation of Li$^+$ (Supplementary Fig. 11).

**Finite element analysis simulation**. Numerical simulations using an in-house Isogeometric Analysis (IGA) code were used to calculate the actuating degree induced by the pseudocapacitive lattice contraction and further verify the main contribution for actuation. The deformation of the dual-responsive film was modeled using the thin shell Kirchhoff−Love theory according to the first principle calculations described above. The variational formulation is stated as follows. Find the displacement of the

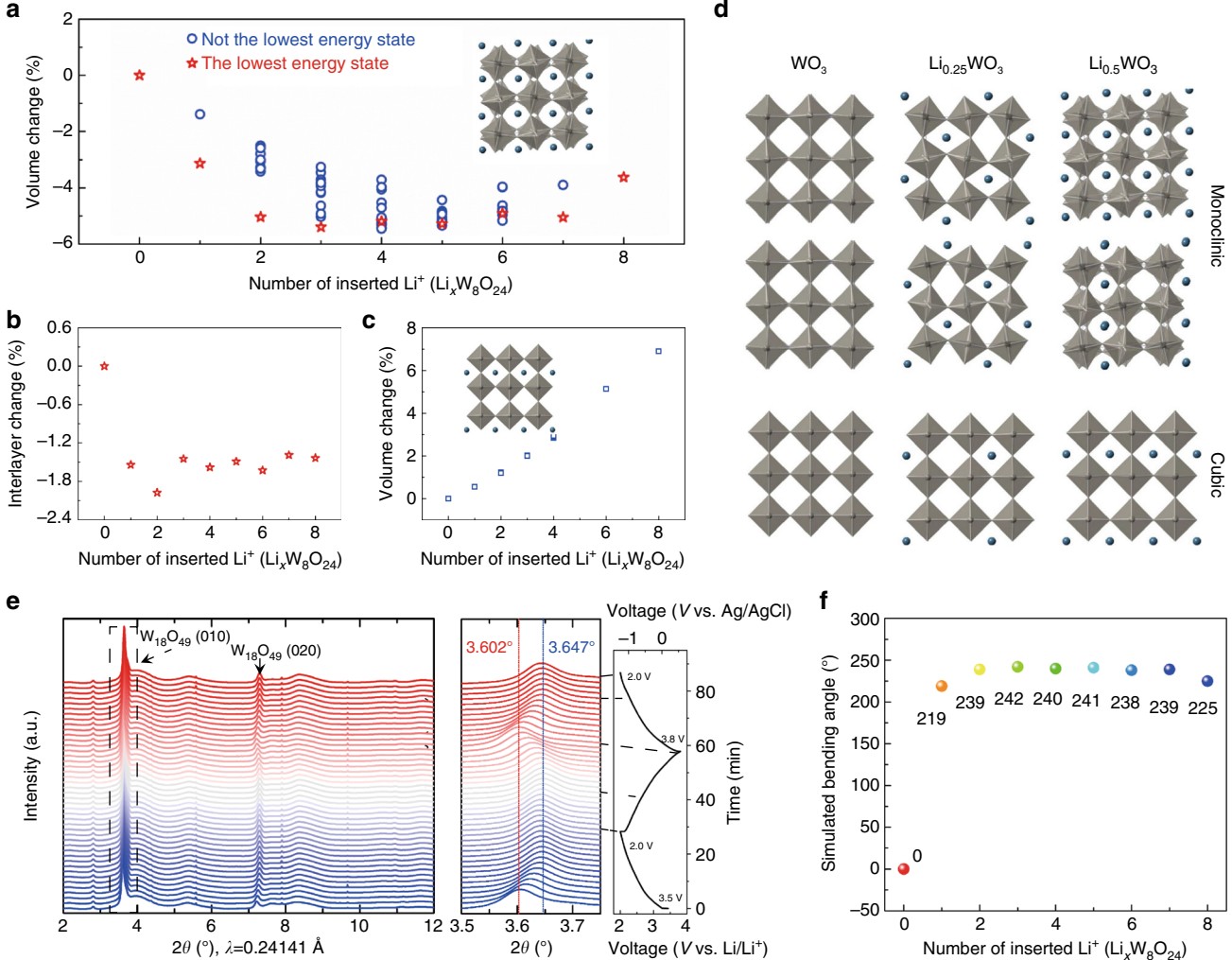

**Fig. 3** The verification for electrochemical actuation of W18O49 nanowires. Volume (**a**) and interlayer (**b**) changes as a function of the number of Li$^+$ inserted into monoclinic WO$_3$; **c** Volume change as a function of the number of Li$^+$ inserted into cubic WO$_3$. **d** The corresponding structural evolution during the Li$^+$ intercalation process; **e** In situ synchrotron X-ray diffraction of W$_{18}$O$_{49}$NWs during lithiation/delithiation processes. **f** Simulated bending angles versus the number of inserted Li$^+$

middle surface of the dual-responsive film $d$, such that $\forall w$

$$
\int_\Gamma^A w\rho^* h^* \left(\frac{\partial^2 d}{\partial t^2} - g\right)\mathrm{d}\Gamma + \int_\Gamma^A \delta\epsilon \cdot (A^*\epsilon + B^*\kappa)\mathrm{d}\Gamma + \\
\int_\Gamma^A \delta\kappa \cdot (B^*\epsilon + D^*\kappa)\mathrm{d}\Gamma = 0
\tag{3}
$$

where w is the testing function, **g** is the gravitational acceleration, $h^*$ is the total thickness of the dual-responsive film, and $\rho^*$, $A^*$, $B^*$, $D^*$ are the homogenized density, membrane, coupling, and bending stiffness matrices, respectively. IGA discretization using quadratic non-uniform rational b-splines (NURBS) basis is adopted to solve the above structural mechanics equations. The final IGA mesh has 25×25 quadratic NURBS elements. As shown in Fig. 3f, after taking the thickness and mechanical properties of each individual layer into account, the maximum simulated bending angle can reach as large as 242° at the condition of three inserted Li$^+$ per W$_8$O$_{24}$ supercell. Notably, this maximum simulated bending angle is quite similar to same as the maximum bending angle of 238° of the dual-responsive films, which indicates that the pseudocapacitive lattice contraction/expansion is the main contribution for the extreme actuation.

**The preparation and performances of air-working actuators.** Bio-inspired actuation materials, also called artificial muscles, were usually used to assemble non-water IPMC actuators. In recent decades, these IPMC actuators have attracted great attention for their hiNgh strain under low voltage stimulation, air-working capability, and fast actuating speed. They have exhibited promising application potentials in intelligent robots, biomedical devices, and micro-electro-mechanical systems. Due to the rapid development of some carbon nanomaterials with the high specific surface area, IPMC actuators are predominantly based on the nonfaradaic reaction mechanism at present.

Based on the intercalation/deintercalation (pseudocapacitive) actuation mechanism, W$_{18}$O$_{49}$NWs were used to construct a solid-state symmetric pseudocapacitive IPMC actuator with polymer gel electrolyte as the ionic storage layer laminated by two pieces of single-wall carbon nanotube (SWCNT)/W$_{18}$O$_{49}$NW composite films (Fig. 4a). As shown in Supplementary Fig. 12, the SWCNTs and W$_{18}$O$_{49}$NWs in the composite films are highly interconnected to form an interpenetrated nanowire network. For our pseudocapacitive IPMC actuators, the Li$^+$ and ClO$_4^-$ are the two mobile ions. However, as with IPMCs, cations (i.e., Li$^+$) are the dominant active ions for our pseudocapacitive IPMC. This is due to the following two reasons: (1) only Li$^+$ can intercalate/de-intercalate

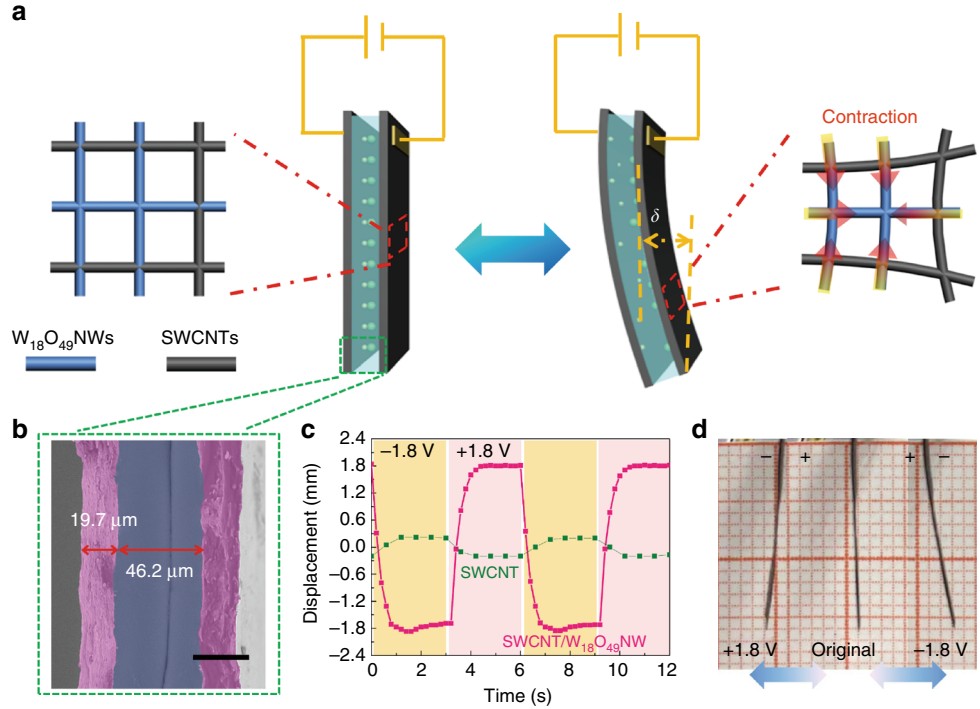

**Fig. 4** The structure and performance of pseudocapacitive ionic polymer metal composite actuators. **a** Schematic illustrations of the assembled configuration and electrochemical actuating mechanism of a pseudocapacitive ionic polymer metal composite (IPMC) actuator; **b** Typical cross-sectional scanning electron microscopy (SEM) image of a pseudocapacitive IPMC actuator (Scale bar: 30 μm); **c** Respective bending displacements of the pseudocapacitive IPMC and pure single-wall carbon nanotubes (SWCNTs)-based IPMC actuators measured between ± 1.8 V bias; **d** The corresponding digital photographs of the pseudocapacitive IPMC actuator under the positive voltage (left), original state (middle) and negative voltage (right), respectively

into the $W_{18}O_{49}$NWs, which is supported by the cyclic voltammetry (CV) curves with only one pair of obvious redox peaks and the obvious peak shift in operando XRD (corresponding to the redox reaction between $Li^+$ and $W_{18}O_{49}$NWs). (2) the diffusion and transport of $Li^+$ are more rapid than bulky perchlorate anions in the gel electrolyte. Therefore, with the intercalation of $Li^+$ into $W_{18}O_{49}$NWs, the spring-like $W_{18}O_{49}$NWs are contracted and become shorter, which results in a decrease of pore size in interpenetrated nanowire networks and the bending of actuator towards to the contracted electrode (Fig. 4a).

As observed in Fig. 4b, the porous SWCNT/$W_{18}O_{49}$NW electrode with high electric conductivity (1590 S cm$^{-1}$) and a thickness of 19.7 μm exhibits good interlayer adhesion with the gel electrolyte, which leads to low interfacial resistance and good ion diffusion between the electrolyte and electrode material. As a result, the maximum bending displacement of pseudocapacitive IPMC actuator reaches ±1.83 mm within 1.4 s under the constant voltage of ±1.8 V (Fig. 4c, d and Supplementary Movie 2) corresponding to the maximum strain (0.12%) and curvature (0.141 cm$^{-1}$) (Supplementary Note 3). As listed in Supplementary Table 2, although the maximum displacement, strain, and curvature are lower than those of many reports in the trilayer configuration, the response speed is much faster than those of the nonfaradaic-reaction-based reduced graphene oxide (rGO) and rGO/multi-wall carbon nanotube (MWCNT) actuators (typically ranging from ~40 to ~500 s)[30,47] and even comparable to best results of illumination-, heat-, moisture- or chemicals-driven actuators[31–34]. Here, three possible reasons are concluded for such outstanding response speed. (1) Compared with the cations of the ionic liquid, Li ions have a much smaller ionic radius (0.72 Å), which is beneficial to electrolyte ions diffusion. (2) There are many three-dimensional interconnected porous channels with the large specific surface area. These channels

allow highly efficient ion migration into $W_{18}O_{49}$ active sites, thereby generating fast and reversible redox reactions. (3) The SWCNTs in the composite electrodes are highly interconnected, which results in fast electron transfer and reduction of the internal resistance. Due to obvious actuating effect and fast response speed, the pseudocapacitive IPMC actuators show great potential to extend application fields of IPMC actuators and can form good complement with nonfaradaic IPMC actuators to improve the comprehensive performance via hybrid structures.

As shown in Fig. 4c and Supplementary Fig. 13, an SWCNTs-based IPMC actuator was prepared using the same assembly method with two pure SWCNT films (ca. 22.0-μm-thick) as two electrodes and was measured as a control experiment. As expected, the SWCNT actuator only exhibit a slight movement at the end of the actuator (only ~0.2 mm) under ±1.8 V. Moreover, the actuating direction of SWCNT actuator is opposite to that of the pseudocapacitive IPMC actuators, clearly indicating that the actuation of pseudocapacitive IPMC actuators is contributed by the $Li^+$ intercalation induced lattice contraction/recovery of $W_{18}O_{49}$NWs.

## Discussion

In previous literature, there is some ambiguity regarding pseudocapacitive actuation in the inorganic materials. For example, Baughman and colleagues reported a $V_2O_5$ nanofiber sheet actuator and stated that the dominant reason for the sheet actuation is probably conversion of 0.495 Å of $V^{5+}$ ions to 0.86 Å of $V^{4+}$ ions during the electrochemical reaction[48]. Cheng and Ngan reported electrochemical actuation of NiO nanohoneycombs[18]. They suggested that the expansion strain might arise from volume expansion due to the reversible reaction between

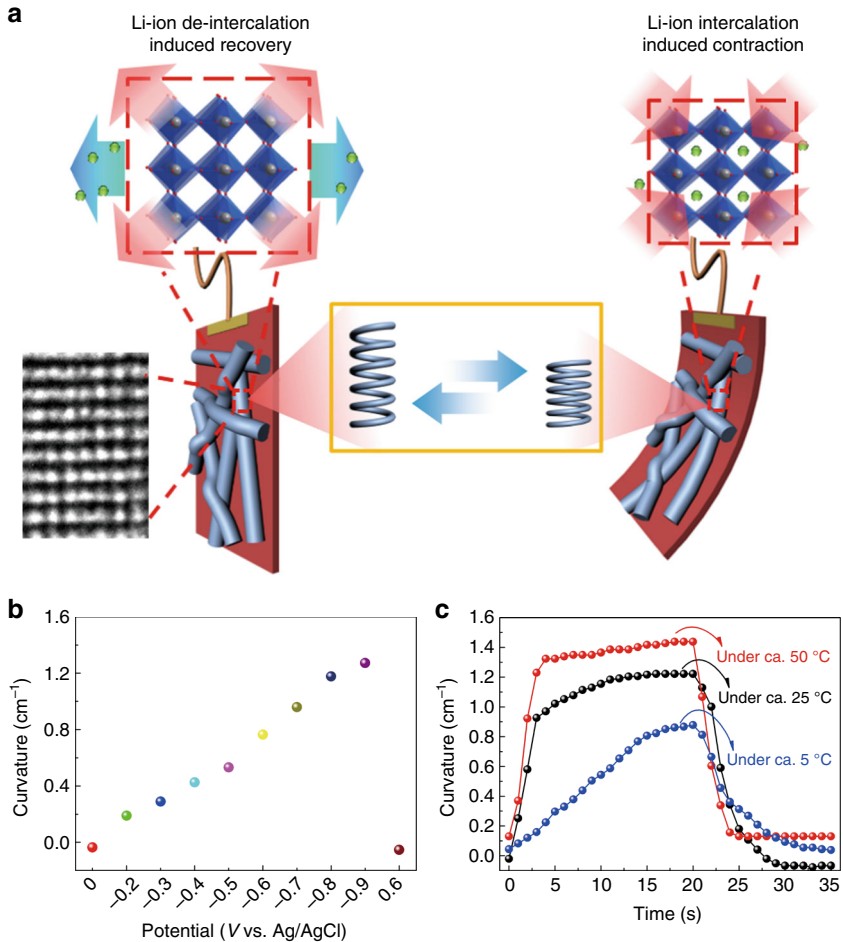

**Fig. 5** The mechanism and tunability of pseudocapacitive actuation. **a** Schematic illustration of actuating mechanism of the dual-responsive film in 1 M LiClO$_4$/ propylene carbonate (PC) electrolyte. **b** Maximum curvatures of the dual-responsive film as function of the applied potentials. **c** Deformation responses of the dual-responsive film measured at +0.6 and −0.9 V bias under different temperatures (~50, ~25 and ~5 °C)

NiO and NiOOH or due to the lattice mismatching between Ni$^{2+}$ and Ni$^{3+}$ but did not provide direct evidence for this. In addition, their actuating performances were much smaller than those of our dual-responsive films and could not be observed clearly by the naked eye. Here, deviating from previous explanations, a new mechanism is elaborated for the unprecedented W$_{18}$O$_{49}$NW-based actuating effect. As illustrated in Fig. 5a, before ion intercalation, the original W$_{18}$O$_{49}$NWs maintain the lattice spacing of 0.38 nm and show a regularly stacked layered crystal structure along the direction of (010). When the Li ions intercalate into the interlayers of W$_{18}$O$_{49}$NWs, the electrostatic interaction between the positive Li ions and polar atoms in the crystal will lead to the lattice contraction and the interlayer spacing decrease. Therefore, the macroscopic deformation is realized by nanostructure changes at the atomic level. In this process, each W$_{18}$O$_{49}$NW is just like an "inorganic EC spring", which will contract and store chemical energy under the stimulation of an electrical field. As the result of this strain difference between the PET film and W$_{18}$O$_{49}$NW network, the composite films will bend.

To thoroughly probe the pseudocapacitive actuation during Li$^+$ intercalation/de-intercalation, we have demented the actuating effect of the dual-responsive films under different applied potentials (Supplementary Fig. 14a), high temperature (Supplementary Fig. 14b), low temperature (Supplementary Fig. 14c), and different shapes (Supplementary Fig. 15), respectively.

First, the multiple potential step method was used to apply different constant potentials which were gradually increased and

maintained for 10 s under each potential. As shown in Fig. 5b and Supplementary Fig. 14d, with the increase of potentials from −0.2 V to −0.9 V, the maximum bending angle and curvature at each potential increased to 32° for −0.2 V, and 215° for −0.9 V. This is mainly because of the intercalation of more Li$^+$ into the W$_{18}$O$_{49}$NWs and further contraction of the lattice spacing with increasing potential. As shown in the operando X-ray synchrotron diffraction of W$_{18}$O$_{49}$NWs (Fig. 3e), the lattice spacing will contract obviously even under a small potential (i.e. ca. −0.2 V vs. Ag/AgCl) and will reach the maximum contraction at ca. 2.2 V, i.e. ca. −1.05 V vs. Ag/AgCl, which is consistent with the bending angle evolution.

In addition, the dual-responsive films were measured at different temperatures. As shown in Fig. 5c and Supplementary Fig. 14e, at the lower temperature (ca. 5 °C), the bending resNponses is remarkably slower than that at ca. 25 °C and maximum bending angle and curvature are also smaller mainly due to the poorer ionic diffusion. The diffusion limitation is supported by the decline of ionic conductivity of the LiClO$_4$/PC electrolyte from 6.96 mS cm$^{-1}$ at ca. 25 °C to 5.15 mS cm$^{-1}$ at ca. 5 °C. At a higher temperature (ca. 50 °C), with the improvement of the ionic conductivity of the electrolyte to 8.96 mS cm$^{-1}$, the dual-responsive films show shorter response times (less than 4 s) and slightly enhanced bending angle range of 25~276 °. These results not only explain that the actuation process is determined by the ion diffusion in both liquid and solid phases, but also show the tunability of the dual-responsive performance under the control of external stimuli.

Third, we trimmed the dual-responsive films into specific shapes, such as triangle, rectangle and circular shapes, to study their actuation capability. As shown in Supplementary Fig. 15, when the voltage was applied, the bending deformation occurred at each corner of the triangle soaked in the electrolyte. For rectangular films, they will bend from both left and right sides (short edges), which is caused by the simultaneous contraction of the four corners. This bending deformation is similar with previously demonstrated rectangular dual-responsive films which also bend from the short edge at the bottom. To demonstrate the bending actuation from more directions, we prepared a circular dual-responsive film and cut it into three parts along the red dashed lines (as shown in the bottom of Supplementary Fig. 15). As expected, the three cut parts bent to the center of circular film, when voltage was applied.

In summary, a bio-inspired flexible EC/actuating dual-responsive film was prepared via the construction of AgNW/$W_{18}O_{49}$NW bilayer networks. These films demonstrate synchronism in pseudocapacitance-induced high-performance EC/actuation. The EC/actuating active material was confirmed as single component, i.e. $W_{18}O_{49}$NWs, via control experiments. More significantly, reversible deformation mechanism of $W_{18}O_{49}$NWs was elucidated as the pseudocapacitive lattice contraction/recovery, which was verified through in-situ synchrotron X-ray diffraction, first principles calculations, numerical simulations, and a series of ex situ structural and elemental characterizations. As an extensive application of pseudocapacitive actuation, an unconventional pseudocapacitive IPMC actuator was constructed based on $W_{18}O_{49}$NWs and demonstrated an obvious displacement. Therefore, this unconventional pseudocapacitive actuating mechanism not only contributes to the technologically relevant electricity-driven color/shape dual-response phenomenon, but also offers the basis for the development of new multifunctional actuators.

Although novel phenomena and systematical analysis have been demonstrated in current work, this pseudocapacitive dual-responsive behavior still needs further comprehensive study. For insNtance, similar to the lattice change in this work, the lattice spacing of some other pseudocapacitive materials will also contract/expand during the ion intercalation/de-intercalation. Therefore, this reversible pseudocapacitive actuation can endow these pseudocapacitive materials with stable actuation performance via the similar lattice structure change and can spur their development in actuating application fields. In addition, according to the XRD and HRTEM characterizations, synthesized $W_{18}O_{49}$NWs mainly grow along [010] direction. During the Li$^+$ intercalation, the crystal lattice contraction also occurs along the direction of [010], which mainly contributes to the macroscopic actuation. Therefore, it is reasonable to hypothesize that orientation of the $W_{18}O_{49}$NWs can augment the strain (i.e. contraction) in the direction along the crystallographic b-axis (perpendicular to the layers in $W_{18}O_{49}$) and highly improve the bending angles. Moreover, we can reduce and even eliminate the strains from other directions, and thus significantly improve the controllability of the bending directions of dual-responsive films. At last, a lot of conducting polymer actuators also exhibit promising actuating performance based on the similar electrochemical mechanism. According to the better elasticity modulus and mechanical flexibility, hybridizing conducting polymer with inorganic actuating materials could be a promising approach to increasing the mechanical flexibility and actuation capability.

## Methods

**Synthesis of $W_{18}O_{49}$ nanowires and Aa-PDA complex**. $W_{18}O_{49}$NWs were synthesized according to a previous report with slight modification[25]. Briefly, 56

mL of transparent yellow ethanol solution containing 0.7 mg mL$^{-1}$ of WCl$_6$ and 4 µg mL$^{-1}$ of polyvinyl pyrrolidone was added into a 70 mL Teflon-lined stainless-steel autoclave. Then, the autoclave was sealed and maintained at 180 °C for 24 h. Then, the product was centrifuged and washed with absolute ethanol. For the comparison of Aa-PDA complex, dopamine hydrochloride (0.125 g) and alginic acid (0.725 g) were quickly added into 50 mL of ammonia solution and reacted for 12 h[49]. After that, 100 mL of absolute ethanol was added into the solution, which caused the complex to precipitate and the precipitate was centrifuged and washed.

**Preparation of the dual-responsive films**. Highly flexible dual-responsive films were prepared according to our previous work with a little modification for performance improvement[19]. Briefly, the Aa–PDA and concentrated AgNW dispersion (5 mg mL$^{-1}$) were added to an ethanol/water solvent mixture (volume ratio of 1:1) to give a dispersion (dispersion A) containing 0.08 mg mL$^{-1}$ of Aa-PDA and 0.65 mg mL$^{-1}$ of AgNWs. A second dispersion (dispersion B) with 0.18 mg mL$^{-1}$ of Aa-PDA and 0.94 mg mL$^{-1}$ of $W_{18}O_{49}$NWs were prepared by adding Aa-PDA and $W_{18}O_{49}$NWs into the ethanol/water solvent. The commercial available PEDOT:PSS$_{(aq)}$ was diluted using ethanol at a 1:4 v/v ratio. Dispersion A, the diluted PEDOT:PSS solution (0.33 mL cm$^{-2}$) and dispersion B (0.8 mL cm$^{-2}$) were successively sprayed onto an ultrathin PET film (ca. 15 mm thick) using a U-star airbrush. Here, the transparent conductive films with sheet resistance of ca. 9.0 Ohm sq$^{-1}$ were used to prepare the dual-responsive films.

**Electrodeposition of amorphous WO$_3$**. Tungsten powder (9.192 g) was dissolved in 100 mL of H$_2$O$_2$ solution (30%) and stirred in an ice water for 30 min. Then, 5 mL of ethanol and 20 mL of water were Nadded and the solution was refluxed at 70 °C for 12 h to form a yellow sol. The electrochemical deposition was performed at −0.7 V for 100 s using a three-electrode system.

**Assembly of the air-working actuators**. 7.5 mg of SWCNTs, 20 mg of sodium dodecylbenzene sulfonate (SDBS) and 50 mL of water were mixed and sonicated to form an SWCNT dispersion. 13 mL of $W_{18}O_{49}$NW dispersion (1.9 mg mL$^{-1}$) was added into the SWCNT dispersion. Then, the mixed dispersion was filtrated and washed to obtain SWCNT/$W_{18}O_{49}$NW composite films. For preparing pure SWCNT films, 60 mL of dispersion only containing 15 mg of SWCNTs and 45 mg of SDBS was used to filtrate. Two pieces of as-obtained composite films with the size of 20 × 3 mm$^2$ were used as the work and counter electrodes, respectively. The polymer gel electrolyte (GPE) was blade-coated on one composite film. After drying it at room temperature for 30 min, another composite film was placed on the GPE-coated composite film to assemble the electrochemical actuators. The pure SWCNT based electrochemical actuators were assembled with SWCNT films as electrodes using the same method. Here, the GPE contained 20 mL of 1 M LiClO$_4$ in PC solution and 17 wt.% PMMA (relative to the LiClO$_4$/PC solution) in addition to 15 mL of acetone.

**Characterization and measurements**. The X-ray diffraction (XRD) patterns were measured using the X-ray diffractometer (D/max 2550 V, Rigaku, Japan, Cu K$_\alpha$ ($\lambda$ = 0.154 nm) radiation at 40 kV and 200 mA). The morphology of the samples was characterized by field emission scanning electron microscopy (FE-SEM, S-4800, Hitachi, Japan). High-resolution transmission electron microscopy (HRTEM) images were obtained using a JEM 2100 F (JEOL, Tokyo, Japan) operating at 200 kV. The sheet resistance was measured using a four-point probe system (MCP-T360, Mitsubishi Chemical, Japan). The transmission spectra of the as-prepared electrodes were measured using a UV-vis spectrophotometer (Lambda 950, Perkin Elmer, Waltham, MA, USA). Cyclic voltammetry and multiple potential step measurements were performed using an electrochemical workstation (CHI760D, Shanghai Chenhua Instruments, China) and via a three-electrode system in a 1.0 M lithium perchlorate (LiClO$_4$)/propylene carbonate (PC) solution. The electrochemical experiments were operated at a relatively constant temperature (~25 °C)/humidity (~40%) environment. The effective length of the dual-responsive film (namely, the length under the solution) and pseudocapacitive IPMC used for actuating measurements, was around 3.2 cm and 1.6 cm, respectively.

**Operando synchrotron X-ray diffraction**. The in situ XRD measurement was performed at the Advanced Photon Source at Argonne National Laboratory. Electrodes suitable for the X-ray scattering measurements were prepared by mixing $W_{18}O_{49}$NWs with carbon (TIMCAL, SuperP) and PTFE (Sigma, free-flowing powder, < 1 µm) in a 5:4:1 mass ratio in an agate mortar and pestle. The composite powder was cold pressed in a 10 mm diameter stainless steel die to 1 MPa. The resulting free-standing pellet, with an active material loading of 11.6 mg·cm$^{-2}$, was placed into a purpose-built two-electrode AMPIX cell with a lithium counter electrode, glass fiber separator (Whatman)[50], and electrolyte–solvent system of 1.0 M LiClO$_4$ in propylene carbonate (PC). The cell was discharged (lithiation of $W_{18}O_{49}$) from open-circuit voltage to 2.0 V vs. Li$^+$/Li then charged (delithiation of $W_{18}O_{49}$) to 3.8 V vs. Li$^+$/Li. The applied current density for both stages of the galvanostatic experiment was 100 mA·g$^{-1}$. Diffraction measurements were carried out in transmission geometry at beamline 17BM-B with an area detector. Each diffraction image consisted of 10 subframes collected for 1 s each. Thus, each 10 s image corresponds to averaging over about 0.002 electrons transferred per tungsten

in $W_{18}O_{49}$. Two-dimensional diffraction rings were integrated into conventional 1D diffraction patterns in GSAS-II[51], which was then used for further data analysis.

The number of intercalated $Li^+$ can be calculated according to the following equation[19],

$$n = \frac{CMV}{F}$$

where $n$ is also the number of the electrons transferred via the redox reaction, $F$ is Faraday's constant, $V$ is the potential window, $M$ is the molecular weight, and $C$ is the capacitance of $W_{18}O_{49}$NWs.

**First principles calculations**. For a cubic cell, a 2×2×2 supercell was constructed; monoclinic $W_8O_{24}$ could be regarded as $2 \times 2 \times 2$ supercell of the cubic $WO_3$ as well. All the compound energies were calculated by density functional theory (DFT) within the Perdew-Burke-Ernzerhof parametrization of the generalized gradient approximation as implemented in the Vienna Ab Initio Simulation (VASP) code. Spin-polarized calculations were performed with ferromagnetic ordering. The strong correlation effect of the transition metals is addressed with the Hubbard U correction (with a U of 6.2 eV for W) to DFT (GGA+U). During the geometry relaxation, both the volume and the shape of the supercell were optimized.

**Finite element analysis simulation**. The deformation of the dual-responsive film is modeled using the thin shell Kirchhoff-Love (KL) theory. Here, for facilitating the simulation, it is assumed that the BNN is a uniform active layer with a thickness of 195 nm and is just consisting of pure $W_{18}O_{49}$NWs due to the much lower loading of AgNWs compared with $W_{18}O_{49}$NWs. The combination of KL theory and IGA are quite suitable for the current applications, based on the two following reasons. First, the thickness of the dual-responsive film is significantly smaller than the other two dimensions, making the thin shell KL theory applicable. Second, as a higher order method, the accuracy of IGA per degree of freedom is much better than that of standard finite element method (FEM). IGA is first proposed in a reported work[52]. Since its conception, IGA has been widely used in fluid mechanics, solid mechanics and structural mechanics simulations[53–57], showing superior performance.

## Data availability
The data that support the findings of this study are available from the corresponding authors upon reasonable request.

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

## Acknowledgements

We gratefully acknowledge the financial support of the NSF of China (51672043), MOE of China (111-2-04, IRT_16R13), STC of Shanghai (16JC1400700, 16XD1400100), SMEC (2017-01-07-00-03-E00055), and Eastern Scholar. G.W. also gratefully acknowledge the mentorship, support, and valuable discussion from Prof. Elsa Reichmanis at Georgia Institute of Technology.

## Author contributions

K.L. and Y.S. contributed equally to this work. K.L., G.W. and H.W. conceived and designed the experiments. K.L., H.F. and X.L. prepared the materials and assembled the devices. K.J.G., Y.S., H.Y., J.L. and B.B. performed the in-situ XRD characterization and analysis. K.L., H.F. and W.H. carried out the electrochemical and morphological characterization and analysis. Z.L. and J.H.Y. performed the DFT calculation and finite element analysis simulation. C.H., Q.Z., J.S.Y. and Y.L. assisted with the analysis of experimental results. K.L., Z.L., J.H.Y., G.W., Y.S. and H.W. co-wrote the paper.

## Additional information

**Competing interests:** The authors declare no competing interests.

