## [Peer Review File · Nature Communications]

Reviewers' comments:

Reviewer #1 (Remarks to the Author):

In this manuscript, Li reported a flexible film with fast respond reversible electrochromic/actuating dual-responsive phenomena. The hypothesis that insect of li ion into W18P49NWs networks was considered as the main reason for this conversion/recovery. The most concern about this is it lack direct in-situ evident such as XPS and in-situ characteristics to support this hypothesis. Overall, this manuscript requires further revision and the revision is more suit to a specific journal. Here are some specific concerns about this manuscript.

1. In page 5 and line 110, how to get the ratio of more than 350 from Figure S1b?
2. In page 6 and line 135, page 17 and line 386 there should be space between the digital and unit.
3. In page 7 and line 147, page 9 and line 198, Figure should be plural.
4. DFT stated that the fully lithiated phase LiWO₃ is in the perovskites, is there any experimental result to support this simulation result?
5. In the methods, all ml or mL should be kept consistent in the manuscript.
6. In Page 9 line 197, If the cations at 0.6V were considered as the dominate reason for the films bending toe the left, why under negative volume it does not bind to the right? What are the performances of the film under other different voltage? Does the binding angles have a mathematic relations vs voltage? Will this bending affected by different humidity or temperature? Those data could help in-depth understanding the Li ion insect/retraction process.
7. In page 11 and line 240, under the DFT calculation, the authors claim that up to 5.4% volume and 2.0% interlayer contraction were observed under the first principle simulation. It is not convincing to support the experiments results that bending of the films could be up to 250%? Will the thickness or the shape of the film lead to the changes of the torsion angles? The authors should claim this.
8. In page 17 and line 347, correct the citation style.
9. In page 17 and line 387-382, long and confused sentence.
10. In page 17 and line 375, line 388, how much amount of ethanol or water was added?
11. There is no structure or element chemical status characterization results to support the effects after adding Li ion.

Reviewer #2 (Remarks to the Author):

What are the major claims of the paper?

The authors describe an innovative electroactive device presenting synchronous color changes (electrochromic or EC) and actuation based on layered architecture combining PET as passive layer and inorganic W18O₄₉ nanowires sheet as electrochemically active layer. When submitted in electrolytic solution to a redox process by electrochemical stimulation, the W18O₄₉ layer is presenting reversible and controllable color changes (as previously described for such material) but also simultaneous volume changes due to lattice contraction/expansion. The resulting displacement of this layered device is a bending motion. A theoretical model is also described to explain the unusual volume change, i.e. contraction during Li⁺ insertion.

Are the claims novel? If not, please identify the major papers that compromise novelty

The claims are novel since no synchronous color change and actuation has been described before for electroactive materials and the model proposes an interesting mechanism to explain these

unusual volumes changes. Moreover the individual performances (EC and actuation) are both relatively good when compared to the literature

Will the paper be of interest to others in the field? Will the paper influence thinking in the field? Considering the proposed electromechanical model this work could indeed be useful to other research groups working on such electroactive metal oxide or more generally in the field of polymer actuators.

Are the claims convincing? If not, what further evidence is needed?

The claims lack of evidences especially since the complex composition of the electroactive layer (W18O49 but also Ag nanowires and PEDOT:PSS) may induce complex mechanism. Indeed Ag can usually be electrochemically active in the potential range used for this study as well as PEDOT:PSS.

Are there other experiments that would strengthen the paper further? How much would they improve it, and how difficult are they likely to be?

Since the pseudocapacitive behavior of this material is a central element of the paper, cyclic voltammetries of device and blanks (Ag NWs, Ag NWs + PEDOT:PSS). It is surprising to not find any CV since experimental conditions of cyclic voltammetry are described in the experimental section.

Are the claims appropriately discussed in the context of previous literature?

Claims are moderately well discussed in the manuscript. On the one side, discussion and comparison with other inorganic based devices seems properly done as well as proposed mechanism. However, comparison and comparison methods with other electroactive devices seem to lack of knowledge of literature. First they claim that the bending angle is higher than almost all the electrochemical actuators (p7). However the bending angle is related to the performance of the electroactive layer but also to the thickness of the substrate. Thin substrate promotes larger angles for the same deformation of the layer. Then, to compare angles as proof of performance is meaningless. Curvature of the beam can be a more efficient way to compare results but strain of the active layer should be calculated and compared to the literature (Methods to calculated strain of the electrodes: bending beam theory or formula described in T. Sugino, K. Kiyohara, I. Takeuchi, K. Mukai, K. Asaka, *Sensors Actuators B Chem.* 2009, 141, 17). On the same level, bending actuation in open-air is characterized by displacement in mm. However, this value depends strongly on the length of the beam. For a constant bending angle, when the length of sample is increasing, the displacement is increasing also. As a consequence the value is not meaningful and should at least be provided with the length of active beam. And finally, many electroactive polymer devices (conducting polymer actuators, bucky-gel, IPMCs,...) are presenting very large bending motion being based on capacitive or pseudo-capacitive mechanism. Moreover, the authors compare their results to classical IPMCs (ionic polymer metal composites) while IPMCs are by nature capacitive systems. If comparison have to be made, it would be necessary to compare also the devices with conducting polymer based actuators since they are also working with pseudo-capacitive mechanism and are able to be actuated both in electrolytic solution (bilayer configuration) or in open-air (trilayer configuration) exactly as the device described in this work. This comparison should be also done when discussing about speed response of open-air actuator since ECP based actuators have been described to actuate at much higher speed (A. Maziz, C. Plesse, C. Soyer, C. Chevrot, D. Teyssié, E. Cattan, F. Vidal, *Adv. Funct. Mater.* 2014, 24, 4851)

When trilayer devices able to operate in open-air are described, laminated SWCNT are used. However ionic storage layer + SWCNTs are already and widely described in the literature as good bending actuators and it would be critical to demonstrate the influence of W18O49 on performances by describing and comparing actuation results only with SWCNTs.

If the manuscript is unacceptable in its present form, does the study seem sufficiently promising that the authors should be encouraged to consider a resubmission in the future?

In my opinion the manuscript in its present form is unacceptable for publishing. However the results and the proposed mechanism are really interesting and could be encouraged for resubmission in the future.

Additional elements:

Is the manuscript clearly written? If not, how could it be made more accessible?

The manuscript is clearly written but organization of figures is annoying for the reader since discussion starts with 3 figures from SI. Moreover it is generally quite difficult to understand what has been done exactly since many parameters are changing in every section. Indeed, the paper lacks of experimental detail (in the experimental section, in the main text and also in figure captions) to be able to fully understand the work and to be able to reproduce the results: electrolyte should be mentioned everywhere since ionic liquid but also LiClO₄/PC are used in different places, thickness of PET should be mentioned, nature of the polymer membrane used for open-air device is not described. On this point, we can mention that these materials are called p-IPMC by the authors but IPMCs are using ionic polymer (polyelectrolyte) like Nafion® with usually only one mobile ion. If membrane used in this work is made of polymer network swollen with electrolyte (which one?) two mobile ions (anion and cations) are available. The question can rise here on the mobile ion mechanism involved in the actuation mechanism. It has been widely described for conducting polymer actuators that depending on the electrolyte, cation dominant or anion dominant motions can take place. When negative potential is applied to the electrode cation insertion can take place (leading to volume expansion) or anions can be expelled (volume contraction). However ionic liquids are usually cation dominant mechanism and LiClO₄/PC is usually anion dominant. This point can be critical since anion expulsion could also explain the observed volume variation. I do believe that the mechanism proposed by the authors is consistent but experimental details should be provided and discussed more precisely.

Another comment: many acronyms are used in the paper and some of them are not (or I missed them) described like TCF or ECF

Responses to the comments

Reviewers' comments:

Reviewer #1:

In this manuscript, Li reported a flexible film with fast respond reversible electrochromic/actuating dual-responsive phenomena. The hypothesis that insect of li ion into W₁₈P₄₉NWs networks was considered as the main reason for this conversion/recovery. The most concern about this is it lack direct in-situ evident such as XPS and in-situ characteristics to support this hypothesis. Overall, this manuscript requires further revision and the revision is more suit to a specific journal. Here are some specific concerns about this manuscript.

We thank the reviewer's valuable comments and suggestions.

Synchronous color (spectral range) change/actuation of materials always attract people's attention and were widely studied due to their significant application prospects in artificial intelligence, visual communication and our highlighted biomimetic dual-stealth camouflage (*Science*, 2002, 297, 983-987; *Science*, 2012, 337, 828-832; *Science*, 2018, 359, 1495-1500; *Nat. Commun.*, 2014, 5, 4899; *Nat. Commun.*, 2018, 9, 590; *Sci. Adv.*, 2018, 4, eaap8203). So far, through the hybrid of multiple materials and integration of different structures, some color/shape dual-responsive devices were achieved, but still have some limitations such as poor controllability, susceptible stimulating conditions, poor coordination or interface incompatibility. Therefore, novel dual-responsive mechanisms and materials still need to be further developed.

In our work, we demonstrated the W₁₈O₄₉ nanowires (W₁₈O₄₉NWs) exhibited reversible, controllable and synchronous color changes (i.e., electrochromic or EC as previously described for such material) and volume changes (i.e., the bending motion) induced by unconventional lattice contraction/expansion during Li-ion insertion/extraction processes. Detailed and precise control experiments, first principles calculations, structure characterization and numerical simulations were deliberately performed to elaborate and verify the pseudocapacitive actuating mechanism. Thus, our work not only innovatively realizes the macroscopic deformation by operating nanostructure change in atomic level, but also provides a clear and general mechanism for other potential electrochemical actuating materials.

According to the reviewer's constructive suggestion about the *in-situ* characterization for the mechanism clarification, the *in-situ* synchrotron X-ray diffraction characterization, electrochemical test and further theoretical investigations have been provided to further elucidate

the proposed mechanism for fast actuation according to the reviewers' suggestions in the revised manuscript. Moreover, appropriate comparisons were supplemented through the calculation of the strains and curvatures.

For the reviewer's suggestion about the *in-situ* X-ray photoelectron spectroscopy (XPS), we need to operate the liquid environment XPS to study the surface structure evolution during the electrochemical test. We contacted and discussed with the experts in this field. The conclusion is that it is very challenge to conduct liquid environment *in-situ* XPS for this work for the following reasons:

First, the XPS is a very surface sensitive technique, which only detects a few nanometers depth. For our electrochemical system, the electrolyte liquid covering the surface of the electrode will affect the detection of the excited electron and bring great difficulty to detect active materials. Moreover, the electrolyte containing many Li ions also interfere with the detection of the intercalated Li ions into $W_{18}O_{49}$ NWs.

Second, it's very hard to achieve a high vacuum level with liquid electrolyte in the system. Even for the ambient pressure XPS system, a vacuum pressure of 50 mbar is still required. In this case, the evaporation of the organic solvent will increase the concentration of the electrolyte and result in deposition of inorganic salt on the surface of active materials, especially due to the very small loading of electrolyte. This will certainly affect the electrochemical performance and surface properties of the active materials.

Third, the impinging X-rays could lead to "beam damage" issues, which may carbonize the organic solvent and contaminate the surface of the active materials. Especially due to the surface sensitivity of XPS, the carbonized products could cover the surface of the electrode, which will definitely affect the detection of $W_{18}O_{49}$ nanowires. The carbonized organic solvent mixed with Li salt will also disturb the detection during the lithiation/delithiation. It should be mentioned that in the XRD measurement, due to the much deeper detective depth of XRD, the very thin carbonized products (amorphous) will not give obvious influence for XRD signals.

All the questions and concerns have been addressed, point-by-point. We believe that the current version with the novelty of the dual-responsive performance and solid mechanistic analysis and verification are suitable for *Nature Communications*.

Q1. In page 5 and line 110, how to get the ratio of more than 350 from Figure S1b?

A: Thank you for your comment. To better clarify your concern and make a more accurate calculation for the aspect ratio, diameters and lengths of $W_{18}O_{49}$ NWs were measured and the diameter and length distribution of sixty randomly selected $W_{18}O_{49}$ NWs were summarized in Figure R1. Here, statistical image analysis software, ImageJ, was utilized to quantify the dimensions of the $W_{18}O_{49}$ NWs. According to the diameter and length distributions, the $W_{18}O_{49}$ NWs display small diameter variation centered around 20 nm and lengths of ca. 12 μ m, which results in a high aspect ratio around 600.

Here, although most $W_{18}O_{49}$ NWs (more than 95%) exhibit the aspect ratio more than 350, we have to admit that our description is not accurate enough and there are still a few of $W_{18}O_{49}$ NWs with the aspect ratio less than 350. Therefore, to elaborate the size of $W_{18}O_{49}$ NWs more accurately and reasonably, revisions have been made to "The $W_{18}O_{49}$ NWs display diameters centered around 20 nm (Fig. 1a and Supplementary Figure 1a) and lengths of ca. 12 μ m (Supplementary Figures 1b,c), which results in an aspect ratio around 600. This high aspect ratio is beneficial to the

improvement of their intrinsic flexibility and formation of highly connected networks.” in page 5 of the revised manuscript.

Figure R1. The diameter (a) and length (b) distributions of as-prepared $W_{18}O_{49}NWs$.

Q2. In page 6 and line 135, page 17 and line 386 there should be space between the digital and unit.

A: We carefully checked the space you mentioned. However, I found there already was a space between the digital and unit in page 6 and line 135, page 17 and line 386.

Q3. In page 7 and line 147, page 9 and line 198, Figure should be plural.

A: Thanks for your careful reading. The Figure has been revised to be plural in the revised manuscript. Additionally, we have double-checked the whole manuscript and corrected other similar mistakes.

Q4. DFT stated that the fully lithiated phase $LiWO_3$ is in the perovskites, is there any experimental result to support this simulation result?

A: Thanks for your careful reading. For the structure of $LiWO_3$, it has been widely studied by previous published papers. For instance, the $LiWO_3$ was categorized as perovskite structure in Table 10.4.1 (Advanced structural inorganic chemistry. Oxford University Press, 2008, 10.). Recently, some other experimental research works investigated the phase evolution of tungsten trioxide upon Li ion intercalation to fully lithiated $LiWO_3$, which also stated the fully lithiated phase $LiWO_3$ is perovskite. (Adv. Funct. Mater. 2011, 21, 2175-2196; ACS Appl. Mater. Interfaces 2016, 8, 24567-24572) In summary, these reports prove that Li ions are positioned at the centers of the perovskite octahedral units, which indicates that WO_3 will form perovskite or distorted perovskite structure at fully lithiated phase.

In addition, according to the *in-situ* synchrotron XRD, only ~0.26 mole of Li ion can intercalate into $W_{18}O_{49}$ ($WO_{2.72}$) to form $Li_{0.26}WO_{2.72}$ (More details can be seen as below), which is far from the fully lithiated state and is impossible to form perovskite structure. Therefore, the first principles calculation that fully lithiated phase $LiWO_3$ is in the perovskites does not influence the discussion of the whole manuscript.

Q5. In the methods, all ml or mL should be kept consistent in the manuscript.

A: Thanks for your suggestion. We have carefully check the manuscript and unified the units of

ml and mL to ml.

Q6. *In Page 9 line 197, if the cations at 0.6V were considered as the dominate reason for the films bending to the left, why under negative volume it does not bend to the right?*

A: It seems there was a misunderstanding regarding our experimental results and description, because we did not describe the 0.6 V of potential can drive the cation and induce the bending to the left. Additionally, it appears that the “negative volume” mentioned in the review refers instead to “negative voltage”. To aid understanding of our experiment result, we have re-organized the related sentence in page 9-10 of the revised manuscript.

In Page 9 line 197, we actually performed a *control experiment* to prove that the mechanism of actuation is faradaic reaction (i.e., pseudocapitance) rather than common nonfaradaic reaction of the nanowire network (previously reported in the literature, such as *Adv. Mater.*, 2015, 27, 7867-7873; *Nat. Commun.*, 2015, 6, 7258). Therefore, EMI^+ and BF_4^- in 1-ethyl-3-methylimidazolium tetrafluoroborate (EMIBF_4) were used as cations and anions.

According to the nonfaradaic reaction mechanism, when a -0.9 V potential was applied to the film, the cations, EMI^+ , with large ionic radius will move into the porous channels of electrodes. With an increase in the number of cations, the electrostatic repulsive interaction among the cations will expand the volume of bilayer nanowire networks (BNNs), leading to the bending of films towards the right. However, in fact, the dual-responsive films still bent to left under the -0.9 V and show obvious EC phenomenon, which is in contrary to theory based on nonfaradaic reaction. These indicate the occurrence of faradaic redox reaction between $\text{W}_{18}\text{O}_{49}$ and EMI^+ as shown in following equation:

According to the equation, if a contrary (namely, positive) voltage was applied, EMI^+ will be extracted from the $\text{W}_{18}\text{O}_{49}\text{NWs}$. The chemical state and crystal structure of $\text{W}_{18}\text{O}_{49}\text{NWs}$ will recover, which leads to the films back to ca. 0° . *During this process, driven by the positive voltage (0.6V), the BF_4^- just adsorb onto the $\text{W}_{18}\text{O}_{49}\text{NWs}$ and only react on the surface of $\text{W}_{18}\text{O}_{49}\text{NWs}$ at most (i.e., no obvious intercalation), which does not significantly affect the crystal structure. This can be further confirmed by only one pair of redox peaks in CV curves corresponding to the redox reaction between $\text{W}_{18}\text{O}_{49}\text{NWs}$ and EMI^+ (Figure R2). Although there exists the electrostatic repulsive force among the BF_4^- , this force cannot lead to the bending of the dual-responsive films possibly due to fewer adsorbed BF_4^- and their smaller size. Therefore, under positive voltage, it just returned to the original position instead of bending to the right.*

Figure R2. CV curves of the dual-response films measured in 1M EMIBF₄/PC electrolyte.

What are the performances of the film under other different voltage? Does the bending angles have a mathematic relations vs voltage? Will this bending affected by different humidity or temperature? Those data could help in-depth understanding the Li ion insertion/retraction process.

Thank you for the reviewer's suggestions. Here, to further probe the pseudocapacitive actuation during Li ion insertion/extraction, we have supplemented the bending angles of the dual-responsive films as a function of different applied potentials (Figure R3a), and different temperatures (Figures R3b, c). We also summarized the relationships of "bending-angles vs. voltage" (Figure R3d) and "temperature dependent actuation" (Figure R3e).

(1) *For the actuation under different potentials*

Firstly, the multiple potential step method was used to apply different constant potentials which were gradually increased and maintained for 10 s at each potential. with the increase of negative potentials from -0.2 to -0.9 V, the more and more Li ions will intercalate into the W₁₈O₄₉NWs, and lead to contraction of the lattice spacing while the potential increases. Therefore, as shown in Figures R3a,d, the maximum bending angles at each potential increased linearly to 215°. We also supplemented the *in situ* X-ray synchrotron diffraction of W₁₈O₄₉NWs (Figure R4). The lattice spacing will contract even under a low potential and will reach the maximum contraction in the ca. 2.2 V vs. Li/Li⁺ (i.e., ca. -1.05 V vs. Ag/AgCl), which is consistent with the bending angle evolution measured under different potentials.

(2) *For the actuation under different temperatures*

In addition, the dual-responsive performance was characterized at different temperatures. As shown in Figure R3e, at the lower temperature (~5 °C), the bending response speed is remarkably slower than that at room-temperature (~25 °C) and maximum bending angle is also reduced to 160°. We believe the decrease of the actuating performance is mainly ascribed to the moderating ionic diffusion, which can be proved by the decline of ionic conductivity of the electrolyte from 6.96 mS/cm (~25 °C) to 5.15 mS/cm (~5 °C). At the higher temperature (~50 °C), an obvious improvement on the ionic conductivity of the electrolyte to 8.96 mS/cm leads to shorter response time (less than 4 s) and enhanced bending angle range of 25°~276°. These results not only explain that the actuation process is determined by the ion diffusion in both liquid and solid phases, but

also show the tunability of the dual-responsive performance under the control of external stimuli.

The related description has been added in Page 19-20 of the revised manuscript.

(3) For the actuation under different humidity

After careful consideration for the influence of humidity, it concluded that humidity is not an appropriate external stimulus for controlling and changing the bending angles. Moreover, it is difficult to be precisely studied. The reason is that the effective part of electrode was soaked in the electrolyte and the electrochemical process was also conducted in the electrolyte. The humidity cannot directly contact or act upon the electrodes. Therefore, it's hard to objectively and precisely verify the relationship between the bending angles and humidity.

Here, it should be mentioned that our electrochemical experiments were operated at a relatively constant temperature (~25 °)/humidity (~40 %) environment to avoid the excessive influences of external conditions and ensure the reliability of the results as far as possible.

Figure R3. Digital photographs of synchronous electrochromic/actuating processes of the dual-responsive films under different applied potentials (a), under the temperature of ca. 50 °C (b) and under the temperature of ca. 5 °C (c), respectively; (d) Corresponding bending angles of the dual-responsive film as function of the applied potentials; (e) Corresponding bending angle response of the dual-responsive film measured at +0.6 and -0.9 V bias under different temperatures. (Scale bars: 3 cm)

Q7. In page 11 and line 240, under the DFT calculation, the authors claim that up to 5.4% volume and 2.0% interlayer contraction were observed under the first principle simulation. It is not convincing to support the experiments results that bending of the films could be up to 250%? Will the thickness or the shape of the film lead to the changes of the torsion angles? The authors should claim this.

A: Thanks for your questions. In our work, besides the DFT calculation, control experiments,

finite element analysis (FEA) simulation and *ex-situ* structure measurements were conducted *in our original manuscript to clarify the volume deformation*. Based on these experiments and calculations, we concluded that the change of layered crystal structure of $W_{18}O_{49}NWs$ mainly contributed to the remarkable actuation during the faradaic (pseudocapacitive) reaction of $W_{18}O_{49}NWs$. Moreover, the FEA simulation was used based on the DFT calculation results. Although only one geometry is displayed and thickness or the size of the film will lead to the change of the bending angles in this paper, the first principle simulation is able to handle complex geometry with different thicknesses. This is represented by the mathematical formulation, in which the homogenized density, membrane, coupling, and bending stiffness matrices, ρ^*, A^*, B^*, D^* , are constructed by integrating over the thickness direction and taking the thickness and properties of each individual layer into account. The resulting simulated angle is consistent with the practical bending angle of the dual-responsive films (238°), indicating that the 5.4% volume and 2.0% interlayer contraction can lead to the large bending angle of 238° .

Additionally, *in the revised manuscript*, we supplemented the cyclic voltammetry (CV) measurement to further prove the relationship between the lattice spacing contraction and film bending. Meanwhile, the *in-situ* X-ray synchrotron diffraction was supplemented to precisely perform the microstructure evolution and support our DFT calculation. The detailed data and statement can be found in the page 10 and 13 of revised manuscript.

Furthermore, we calculated the strains with the measured bending angles. As described in page 8 of the revised manuscript, the maximum strain of the whole active layer (i.e., the AgNWs/ $W_{18}O_{49}NWs$) can reach 1.81%, which is close to the value of DFT calculation (2.0% interlayer contraction). This also testifies our result that 2.0% interlayer contraction can result in the large bending angle of 238° .

Lastly, as the reviewer mentioned, the thickness and size of the films will affect the bending angles. To be honest, some other factors, including intrinsic actuating performance and thickness of active layers, mechanical properties of substrate, could have influence on the bending angles of the films. Compared with the bending angle, curvature and strain are more independent parameters. Therefore, in order to compare our results in a more rational and scientific way, the curvature and strain were calculated (according to the Supplementary Notes 1-3) to compare with the results of other related actuators with similar bilayer configuration and reaction environment as listed in Supplementary Table 1.

Q8. *In page 17 and line 347, correct the citation style.*

A: The citation style has been corrected. In addition, all the citation styles have been double checked thoroughly.

Q9. *In page 17 and line 387-382, long and confused sentence.*

A: We thank the reviewer for helping us to clarify the verbose lines 378-382 in the Methods section. For this concern, we re-organized the sentence to make it more clearly through separating the one sentence into different short sentences and describe these procedures step by step. The revised sentence is provided as following.

“Briefly, the Aa-PDA and concentrated AgNW dispersion (5 mg mL^{-1}) were added to an ethanol/water solvent mixture (volume ratio of 1:1) to give a dispersion (dispersion A) containing 0.08 mg mL^{-1} of Aa-PDA and 0.65 mg mL^{-1} of AgNWs. A second dispersion (dispersion B) with

0.18 mg mL⁻¹ of Aa-PDA and 0.94 mg mL⁻¹ of W₁₈O₄₉NWs were prepared by adding Aa-PDA and W₁₈O₄₉NWs into the ethanol/water solvent. The commercial available PEDOT:PSS_(aq) was diluted using ethanol at a 1:4 v/v ratio. Dispersion A, the diluted PEDOT:PSS solution (0.33 ml cm⁻²) and dispersion B (0.8 ml cm⁻²) were successively sprayed onto an ultrathin PET film (ca. 15 mm thick) using a U-star airbrush.”

Q10. In page 17 and line 375, line 388, how much amount of ethanol or water was added?

No problem.

A: The amount of ethanol or water have been added in page 21 of our revised manuscript.

Q11. There is no structure or element chemical status characterization results to support the effects after adding Li ion.

A: Thank you for your comment. As reviewer suggested, although we performed many *ex-situ* structural characterization, the real-time observation for the structural changes are really needed.

To further reveal crystal structure transformation of W₁₈O₄₉ during the lithiation/delithiation electrochemical processes, we performed the *in-situ* synchrotron XRD. The *in-situ* XRD is a direct and powerful method, which can not only probe the gradual changes of the crystal structure during the lithiation process, but also provide initial potential for inducing the microstructure deformation. The *operando* electrochemical measurement of W₁₈O₄₉ and synchrotron diffraction were performed with high energy X-ray (51.358 keV) for 1.5 cycles corresponding to lithiation-delithiation-lithiation at a current density of 100 mA g⁻¹. It is noted that due to the Li metal counter electrode used for the reference electrode in *in-situ* synchrotron XRD electrochemical cell, the potentials reported here are reported versus Li⁺/Li. As shown in Figure R4, the galvanostatic electrochemistry was conducted from open-circuit voltage (3.5 V) to 2.0 V (vs. Li/Li⁺, i.e., -1.25 V vs. Ag/AgCl) on lithiation and to 3.8 V (vs. Li/Li⁺; i.e., 0.55 V vs. Ag/AgCl) on delithiation. The structure evolution of W₁₈O₄₉ was derived from the *operando* synchrotron diffraction. Based on its nanowire structure and long axis along the *b*-direction, the XRD pattern of W₁₈O₄₉ was dominated by crystallographic (010) and (020) reflections. The (*h*00) reflections are severely broadened due to their nanoscale dimensions, which is consistent with the high-resolution TEM results.

As shown in the enlarged XRD pattern from 2θ= 3.5° to 3.8°, we clearly observed the reversible contraction and extension of crystal lattice face (010) spacing, i.e., the crystallographic *b*-axis (perpendicular to the layers in W₁₈O₄₉), during the electrochemical lithiation/delithiation processes. During the initial lithiation from 3.5 V to 2.23 V (i.e., -1.02 V vs. Ag/AgCl), it shows a gradual right-shift from 3.602° to 3.647°, corresponding to 1.25% structure contraction. Here, the mole ratio of intercalated Li⁺ (relative to the WO_{2.72}, i.e., W₁₈O₄₉) can be calculated to be ~ 0.26 (Li_{0.26}WO_{2.72}), according to the following equation (Small 2017, 13, 1700380),

$$n = \frac{CMV}{F}$$

where *n* is also the number of the electrons transferred *via* the redox reaction, *F* is Faraday's constant, *V* is the potential window, *C* is the capacitance of W₁₈O₄₉NWs and *M* is the molecular weight. This mole ratio is consistent with the calculated result of the required mole ratio of Li⁺ (Li_{0.25}WO₃) for maximum volume decrease. When the voltage continues to decrease to 2.0 V, the peaks do not show an obvious shift and maintain roughly constant, which is also roughly

consistent with the previous DFT calculation results. During the following delithiation and 2nd repeated lithiation process, the crystal lattice spacing of (010) face expanded and contracted, proving this process is reversible and repeatable.

The related description was added in the page 12-13 of the revised manuscript.

Figure R4. *Operando* X-ray synchrotron diffraction of $W_{18}O_{49}NWs$ during lithiation and delithiation processes.

Reviewer #2 (Remarks to the Author):

1. What are the major claims of the paper?

The authors describe an innovative electroactive device presenting synchronous color changes (electrochromic or EC) and actuation based on layered architecture combining PET as passive layer and inorganic W₁₈O₄₉ nanowires sheet as electrochemically active layer. When submitted in electrolytic solution to a redox process by electrochemical stimulation, the W₁₈O₄₉ layer is presenting reversible and controllable color changes (as previously described for such material) but also simultaneous volume changes due to lattice contraction/expansion. The resulting displacement of this layered device is a bending motion. A theoretical model is also described to explain the unusual volume change, i.e. contraction during Li⁺ insertion.

2. Are the claims novel? If not, please identify the major papers that compromise novelty.

The claims are novel since no synchronous color change and actuation has been described before for electroactive materials and the model proposes an interesting mechanism to explain these unusual volumes changes. Moreover the individual performances (EC and actuation) are both relatively good when compared to the literature

3. Will the paper be of interest to others in the field? Will the paper influence thinking in the field?

Considering the proposed electromechanical model this work could indeed be useful to other research groups working on such electroactive metal oxide or more generally in the field of polymer actuators.

A (The response for comments 1-3): We are grateful to Reviewer 2 for recognizing the significance and novelty of our contribution to the scientific community. According to the reviewer's comments as following, we have carefully verified our conclusions, rechecked the data and provided appropriate comparison. In addition, the *in-situ* synchrotron X-ray diffraction characterization and additional experimental and theoretical investigations have been provided to further elucidate the proposed mechanism for the fast actuation. We believe that the current version can be published in *Nature Communications*.

4. Are the claims convincing? If not, what further evidence is needed?

The claims lack of evidences especially since the complex composition of the electroactive layer (W₁₈O₄₉ but also Ag nanowires and PEDOT:PSS) may induce complex mechanism. Indeed Ag can usually be electrochemically active in the potential range used for this study as well as PEDOT:PSS.

A: It's a good question. As the reviewer mentioned, the complex components (W₁₈O₄₉NWs, PEDOT:PSS and AgNWs) in dual-responsive films indeed initially brought difficulty to figure out the main mechanism of electrochemical actuation. Therefore, to find out the main active component and mechanism, we carried out many control experiments, structural measurements, theoretical calculation/simulation in our original manuscript and supplemented the *in-situ* structural and electrochemical characterization to support our conclusion in the revised manuscript. Here, the main results are concluded briefly as following:

(1) The AgNWs/PEDOT:PSS coated on PET substrate was tested under the same electrochemical condition as that used for the dual-responsive film. As shown in Supplementary

Figure 5, there is no deformation or displacement probably due to very small PEDOT:PSS loading and confinement of PET substrate. *Thus, we can confirm the actuation is not caused by the AgNWs and PEDOT:PSS.*

(2) The dual-responsive film was measured in PC electrolyte containing 1 M 1-ethyl-3-methylimidazolium tetrafluoroborate (EMIBF₄) which is an ionic liquid and is usually used in nonfaradaic reaction based actuators. According to the nonfaradaic reaction based actuating mechanism, if the negative voltage is applied, the cations, EMI⁺, with large ionic radius will move into the porous channels of electrodes and adsorb on the nanowires, which lead to volume expansion of BNNs and bending of the films to the right. However, in fact, under the voltage of -0.9 V, the dual-responsive films still bent to left, which is contrary to the theory based on nonfaradaic reaction (Figs. 2a, c). Moreover, obvious EC phenomenon is also observed, indicating the occurrence of faradaic redox reaction between W₁₈O₄₉ and EMI⁺. *These phenomena prove that the faradaic redox reactions of W₁₈O₄₉NWs are responsible for the electrochemical actuation instead of the nonfaradaic mechanism of BNNs.*

(3) Amorphous WO₃ (a-WO₃) was electrodeposited onto the AgNWs/PEDOT:PSS composite film and was also measured in LiClO₄/PC electrolyte. As shown in Supplementary Figure 6d, the AgNWs/PEDOT:PSS/a-WO₃ composite film slightly bent to the right, which is inconsistent with deformation of the dual-responsive film. This is probably due to expansion of a-WO₃ during the insertion of Li ions (*Angew. Chem., Int. Ed.* 2016, 55, 1; *Angew. Chem., Int. Ed.* 2015, 54, 15222). According to the analysis described above, *we can confirm that the deformation of layered crystal structure of W₁₈O₄₉NWs is the key for the remarkable actuation during the faradaic (pseudocapacitance) reaction of W₁₈O₄₉.*

(4) *Ex-situ* structural characterizations (XRD and TEM), DFT calculation and simulation provided strong evidence to prove the lattice contraction/recovery based actuating mechanism during the faradaic redox process.

(5) More importantly, we supplemented the cyclic voltammetry (CV) measurement to further prove the relationship between the lattice spacing contraction and film bending. Meanwhile, the *in-situ* X-ray synchrotron diffraction was supplemented to precisely perform the microstructure evolution and support our DFT calculation. The detailed data and statement can be found in the page 10 and 13 of revised manuscript.

In order to more clearly illustrate our results and conclusions, related sentences were re-organized and the detail electrochemical conditions were added and in page 9, 10 and 13 of the revised manuscript.

5. Are there other experiments that would strengthen the paper further? How much would they improve it, and how difficult are they likely to be?

Since the pseudocapacitive behavior of this material is a central element of the paper, cyclic voltametries of device and blanks (Ag NWs, Ag NWs + PEDOT:PSS). It is surprising to not find any CV since experimental conditions of cyclic voltammetry are described in the experimental section.

A: Very constructive suggestions. As shown Figure R6, we have measured and supplemented the cyclic voltammetry (CV) curves of and *AgNWs/PEDOT:PSS/-W₁₈O₄₉NWs films in 1 M LiClO₄/PC, AgNWs/PEDOT:PSS films in 1 M LiClO₄/PC (for blank) and AgNWs/PEDOT:PSS/-W₁₈O₄₉NWs films in 1 M EMIBF₄/PC (for control), respectively. According*

to these CV curves, related redox reactions and potentials for dual-responsive film and control experiments were described in detail and added in page 10 of revised manuscript and in Supplementary Figure 7 of Supplementary Information, which further support our results about dual-responsive mechanism.

The detailed description and figures are also provided as below:

The cyclic voltammetry curves of different films were measured, compared and studied. As shown in Figure R6a, the dual-responsive film shows typical CV curve (*Nano Lett.*, 2017, 17, 5756-5761; *Nano Lett.*, 2013, 13, 3589-3593; *J. Mater. Chem.*, 2012, 22, 16633-16639; *Sci. Rep.*, 2013, 3, 1936.) and only one pair of redox peaks at the respective potentials of ca. -0.72 and -0.46 V under the scan rate of 5 mV/s. Such redox peaks are contributed from the redox reaction of $W_{18}O_{49}$ during Li^+ intercalation/de-intercalation, indicating that the main active ions in this dual-responsive process are cations (i.e., Li^+ dominant reaction). In contrast, AgNWs/PEDOT:PSS composite films exhibited totally different CV curves with very broad redox peaks (Figure R6b), which are similar with typical CV curves of PEDOT:PSS. Moreover, the current densities of AgNWs/PEDOT:PSS composite films are much lower than those of the dual-responsive film, which also proves very low loading of PEDOT:PSS. For the dual-response films measured in 1M EMIBF₄/PC electrolyte, they possess relatively obvious redox peaks rather than the nearly-rectangular curves (Figure R6c), which further supports the above description about faradaic redox reaction between $W_{18}O_{49}$ and EMI^+ . However, as shown in Figure R6d, under the same scan rate, the current density of the dual-response films measured in 1M EMIBF₄/PC electrolyte are lower than those measured in 1M LiClO₄/PC electrolyte and the positions of redox peaks also shifted to higher potentials (ca. -0.9/0.015 V), due to the moderate solid-state diffusion rate of EMI^+ with its larger ionic radius.

Figure R6. Cyclic voltammetry (CV) curves of the dual-response films (a) and AgNWs/PEDOT:PSS composite films (b) measured in 1M LiClO₄/PC electrolyte; (c) CV curves of the dual-response films measured in 1M EMIBF₄/PC electrolyte; (d) The comparison among the respective CV curves (at scan rate of 5 mV/s) under the above three conditions.

6. Are the claims appropriately discussed in the context of previous literature?

Claims are moderately well discussed in the manuscript. On the one side, discussion and comparison with other inorganic based devices seems properly done as well as proposed mechanism. However, comparison and comparison methods with other electroactive devices seem to lack of knowledge of literature. First they claim that the bending angle is higher than almost all the electrochemical actuators (p7). However the bending angle is related to the performance of the electroactive layer but also to the thickness of the substrate. Thin substrate promotes larger angles for the same deformation of the layer. Then, to compare angles as proof of performance is meaningless. Curvature of the beam can be a more efficient way to compare results but strain of the active layer should be calculated and compared to the literature (Methods to calculate strain of the electrodes: bending beam theory or formula described in T. Sugino, K. Kiyohara, I. Takeuchi, K. Mukai, K. Asaka, *Sensors Actuators B Chem.* 2009, 141, 17).

A: Great suggestions! As the reviewer said, the bending angle is related to many parameters, such as intrinsic actuating performance and thickness of active layers, substrate thickness, mechanical properties of substrate, length of actuators and so on. Therefore, the only parameter of bending angle seems not rational enough to compare the actuating performance with previously published research works.

Compared with the bending angles, curvature and strain are more independent and objective parameters. Therefore, according to the reviewer's suggestion, the curvature and strain were calculated to be 1.22 cm^{-1} and 1.81%, respectively, and were used to compare with the results of other related actuators with similar bilayer configuration and reaction environment as listed in Supplementary Table 1. It can be found that the maximum actuating curvature, strain, and angle were higher than most reported conducting polymer-based results, which is also identical to our original conclusion. Here, the calculating equations were also provided in Note 1 and 2 of Supplementary Information for reviewer's and reader's information.

On the same level, bending actuation in open-air is characterized by displacement in mm. However, this value depends strongly on the length of the beam. For a constant bending angle, when the length of sample is increasing, the displacement is increasing also. As a consequence the value is not meaningful and should at least be provided with the length of active beam.

A: According to your comments, we have revised three main parts to enrich the results and make our comparisons more reasonable in the revised manuscript.

(1) The effective length of the pseudocapacitive IPMC actuators (ca.1.6 cm) was provided in the Methods section of the revised manuscript.

(2) The detailed preparation process including the size of the pseudocapacitive IPMC, components and their proportion of gel electrolyte, and mass ratio between SWCNTs and $\text{W}_{18}\text{O}_{49}\text{NWs}$, have been added in Methods section.

(3) The maximum strain and curvature during actuating process were also calculated by respective 0.12% and 0.141 cm^{-1} according to the equations in Supplementary Note 3. Moreover, these values were listed in Supplementary Table 2 and were comprehensively compared with other related actuators with similar trilayer configuration.

To be honest, the maximum displacement, strain and curvature of current work are not as good as those showed in previously reported the trilayer actuators. However, the response time is indeed shorter than those of most previously reported IPMCs and conducting polymer-based actuators. We believe that the actuating range can be further enhanced in our future work, because the

maximum strain of the active layer is far lower than the value measured in the bilayer configuration. Moreover, the pseudocapacitive IPMC actuators with fast actuating speed are suitable for improving the comprehensive performance of other actuators via hybrid structures.

And finally, many electroactive polymer devices (conducting polymer actuators, bucky-gel, IPMCs,...) are presenting very large bending motion being based on capacitive or pseudo-capacitive mechanism. Moreover, the authors compare their results to classical IPMCs (ionic polymer metal composites) while IPMCs are by nature capacitive systems. If comparison have to be made, it would be necessary to compare also the devices with conducting polymer based actuators since they are also working with pseudo-capacitive mechanism and are able to be actuated both in electrolytic solution (bilayer configuration) or in open-air (trilayer configuration) exactly as the device described in this work. This comparison should be also done when discussing about speed response of open-air actuator since ECP based actuators have been described to actuate at much higher speed (A. Maziz, C. Plesse, C. Soyer, C. Chevrot, D. Teyssié, E. Cattan, F. Vidal, *Adv. Funct. Mater.* 2014, 24, 4851)

A: Thanks for your comment and suggestion. After careful consideration, we indeed found that the comparison between the pseudocapacitive IPMC and typical IPMCs was not enough. Therefore, according to your suggestion, many related results about conducting polymer-based actuators and IPMCs were supplemented in Supplementary Table 2. The comprehensive performances of the pseudocapacitive IPMC actuators are not the best one compared with all the electrochemical actuators which can work in open-air. However, the response time of the pseudocapacitive IPMC is shorter than most values of previously reported IPMCs and conducting polymers based actuators. Therefore, this unconventional actuating material still shows competitive performance in some aspects and this pseudocapacitive actuating mechanism also provides a novel approach to improve the comprehensive performance of other actuators via hybrid structures.

Another concern is that response speed of ECP based open-air actuators in previously reported work (especially in *Adv. Funct. Mater.*, 2014, 24, 4851) was described to be much faster than our pseudocapacitive IPMC. Based on this concern, their frequency is indeed faster than our response time. However, in their work, they used voltage with different frequencies to drive the deformation of actuator. Actually, in these frequencies, the actuator did not reach the maximum deformation or displacement, which could be proved by no formation of obvious horizontal platform after displacement from one side to another side in their displacement-time curves. Moreover, although higher frequency can be used to shorten the response time, higher frequency always leads to the obvious decrease of the actuation amplitude due to insufficient redox reaction/ion diffusion (*J. Mater. Chem. A*, 2014, 2, 16836-16841; *Adv. Mater.*, 2016, 28, 1610-1615; *Nat. Commun.*, 2018, 9, 752). Therefore, the meaning of their frequency or response time is different than that of the response time which is generally defined as the time for a stable maximum displacement.

When trilayer devices able to operate in open-air are described, laminated SWCNT are used. However ionic storage layer + SWCNTs are already and widely described in the literature as good bending actuators and it would be critical to demonstrate the influence of $W_{18}O_{49}$ on performances by describing and comparing actuation results only with SWCNTs.

A: Great suggestion! As shown in Figures R7 and R8, a pure SWCNTs-based IPMC actuator was prepared using the same assembly method and gel electrolyte and their displacement was measured as a control experiment. As expected, the SWCNT actuator did not show obvious deformation except for a slight movement of the end of the actuator (only ~0.2 mm) under ± 1.8 V. Moreover, the actuating direction of SWCNT actuator is opposite to that of the pseudocapacitive IPMC actuators, clearly indicating that the actuation of pseudocapacitive IPMC actuators is contributed by the Li^+ intercalation induced lattice contraction/recovery of $\text{W}_{18}\text{O}_{49}$ NWs instead of the SWCNTs.

Interestingly, we found that the actuating direction of SWCNT actuators was opposite to that of the pseudocapacitive IPMC actuators. A possible explanation concerns the electrolyte. For the assembly of air-operated SWCNT actuators, gel electrolytes (PMMA/PC/LiClO₄) must be used with the polymer as a mechanical framework. However, the polymer component in the gel electrolyte will block the efficient diffusion of ions, which leads to the decrease of ionic conductivity. However, due to smaller ionic radius, Li ions (0.73 Å without solvation) can diffuse in gel polymer electrolyte and insert into the porous electrodes more easily and efficiently compared with ClO₄⁻ (2.40 Å without solvation) with much larger radius (<http://www.wiredchemist.com/chemistry/data/thermochemical-radii-anions>). Therefore, the cation (Li^+) diffusion and movement are dominant in the actuating process of SWCNT actuators. In this situation, when a voltage was applied, many Li^+ will move into negative electrodes quickly and lead to significant expansion of electrodes caused by the electrostatic repulsion among Li^+ . Eventually, SWCNT actuators will be bent to positive electrodes which is opposite to the bending direction of pseudocapacitive IPMC.

Related description has been added in the page 17 of revised manuscript.

Figure R7. Respective bending displacements of the pseudocapacitive IPMC and pure SWCNT-based actuators measured between ± 1.8 V bias.

Figure R8. The corresponding digital photographs of the SWCNT based actuator under the negative voltage (left), original state (middle) and positive voltage (right), respectively.

7. *If the manuscript is unacceptable in its present form, does the study seem sufficiently promising that the authors should be encouraged to consider a resubmission in the future?*

In my opinion the manuscript in its present form is unacceptable for publishing. However the results and the proposed mechanism are really interesting and could be encouraged for resubmission in the future.

A: We are grateful to Reviewer 2 for recognizing the significance and novelty of our contribution to the scientific community. According to your constructive and valuable suggestions/comments, major revisions have been implemented to make the revised version acceptable for publishing.

Additional elements:

8. *Is the manuscript clearly written? If not, how could it be made more accessible?*

The manuscript is clearly written but organization of figures is annoying for the reader since discussion starts with 3 figures from SI. Moreover it is generally quite difficult to understand what has been done exactly since many parameters are changing in every section. Indeed, the paper lacks of experimental detail (in the experimental section, in the main text and also in figure captions) to be able to fully understand the work and to be able to reproduce the results: electrolyte should be mentioned everywhere since ionic liquid but also LiClO₄/PC are used in different places, thickness of PET should be mentioned, nature of the polymer membrane used for open-air device is not described.

A: We appreciate the reviewer's responsibility toward clarify. According to the reviewer's suggestions, we revised our paper in two aspects. (1) The order of sentence was reorganized in pages 5~6 of our revised manuscript. The microstructure of W₁₈O₄₉NWs was firstly elaborated. Then, the dual-responsive film prepared based on the W₁₈O₄₉NWs was described. In this case, the discussion part starts with figures 1 instead of three figures from Supplementary Information. (2) More experimental details were supplemented in the Method section including the assembling process of pseudocapacitive IPMC actuators, preparation of GPE and the precious size of prepared actuators. Moreover, we have appended the key parameters or experimental conditions in figure captions or figures (not only in Method section or text), such as the thickness of substrate,

electrolyte, concentration and applied potential, to ensure the results can be more easily understood and reproduced.

On this point, we can mention that these materials are called p-IPMC by the authors but IPMCs are using ionic polymer (polyelectrolyte) like Nafion® with usually only one mobile ion. If membrane used in this work is made of polymer network swollen with electrolyte (which one?) two mobile ions (anion and cations) are available. The question can rise here on the mobile ion mechanism involved in the actuation mechanism. It has been widely described for conducting polymer actuators that depending on the electrolyte, cation dominant or anion dominant motions can take place. When negative potential is applied to the electrode cation insertion can take place (leading to volume expansion) or anions can be expelled (volume contraction). However ionic liquids are usually cation dominant mechanism and LiClO₄/PC is usually anion dominant. This point can be critical since anion expulsion could also explain the observed volume variation. I do believe that the mechanism proposed by the authors is consistent but experimental details should be provided and discussed more precisely.

A: We very much appreciate your constructive suggestions.

First, we regret that the experiment-IPMC preparation procedures for the pseudocapacitive IPMC were not provided in original manuscript. In Method section of our revised manuscript, the detailed materials and preparation process were supplemented.

Second, as the reviewer said, for the typical or traditional IPMC devices, ionic polymers (polyelectrolyte) with one mobile ions were used as the electrolyte. However, in recent years, with the fast development of IPMC, many researchers used ionic liquids (typical electrolyte with two types of mobile ions: positive and negative) to prepare the electrolyte layer of IPMCs (*Nano Lett.*, 2011, 11, 4636; *Nat. Commun.*, 2015, 6, 7258; *Nat. Commun.*, 2018, 9, 752). Noteworthy, although two types of ions are in the electrolyte, the main active ions used in the IPMCs are positive ions with much larger ionic radius than that of negative ions (i.e., the actuating process is cation dominant). These large positive ions can more easily lead to the expansion of electrodes by means of the electrostatic repulsion among them.

Third, as you described about the conducting polymer actuators, one dominant ion is used for doping/de-doping with conducting polymer as well. This is determined by the redox reaction mechanism of conducting polymers.

Lastly, for the pseudocapacitive IPMC devices, the LiClO₄/PC/PMMA was used as the gel electrolyte, namely, the Li⁺ and ClO₄⁻ are the two types of ions. However, very similar with device structure and dominant ions of IPMCs, the main active ions for the pseudocapacitive IPMC is Li⁺ (i.e., cation dominant). This is because of two reasons: (1) only the Li⁺ can intercalate/de-intercalate into the W₁₈O₄₉NWs, which can be proved by our CV measurement only with one pair of obvious redox peaks and the obvious peak shift in *operando* XRD (indicating the redox reaction between Li⁺ and W₁₈O₄₉NWs) and other previous literature (*Adv. Mater.*, 2017, 29, 1606728; *Angew. Chem. Int. Ed.*, 2014, 53, 11935; *Adv. Mater.*, 28, 10518-10528.). (2) due to its smaller ionic radius, the diffusion and transport of Li⁺ are much easier than anions in the gel electrolyte. Moreover, our control experiment about actuation of pure SWCNT films also prove that anion induced expansion is not the main reason for actuation of our pseudocapacitive IPMC actuators.

A related description was added in the page 15 of revised manuscript.

9. Another comment: many acronyms are used in the paper and some of them are not (or I missed them) described like TCF or ECF

A: According to your comments, the acronyms (like TCF or ECF) have been revised with the full name in the Supplementary Figure 2d of Supplementary Information and double-checked the whole manuscript to avoid similar mistakes.

Reviewers' comments:

Reviewer #1 (Remarks to the Author):

The manuscript has been improved with in-situ XRD and clear mechanism explanations. However, there are few more questions before this manuscript can be accepted.

1. In Figure 2b i would suggest the author mirror the image and let the film bends to left, which would consist with the Figure 2c images.
2. The mechanism of the pseudocapacitive actuation is novel, can the authors provide more quantification discussion to predict the bending angles of the film under different voltages or nanowires directions. So that this model would be more general and guide other polymer actuators design.
3. To further confirm the mechanism, the film should bend different angles vs different film shape (triangle, round shape)? The authors can further test this.

Responses to the referee

Reviewers' comments:

The manuscript has been improved with in-situ XRD and clear mechanism explanations. However, there are few more questions before this manuscript can be accepted.

We are grateful to the reviewer for appreciating the improvement of our last revision and thank the reviewer's additional valuable suggestions. According to the reviewer's comments, we have supplemented some experiments and analysis carefully, and provided some promising prospects for next step work. We believe that the current version can match the requirements for publishing in *Nature Communications*.

Q1. *In Figure 2b, i would suggest the author mirror the image and let the film bends to left, which would consist with the Figure 2c images.*

A: Thanks for the reviewer's comment.

Actually, the Figure 2b is to schematically illustrate the "assumptive" **non-Faradaic** reaction actuation mechanism, which is to distinguish with and further prove the actual **Faradaic** actuation process demonstrated in Figure 2c. Therefore, the bending direction in Figure 2b is inverse with that in Figure 2c. The reason that we put this illustration over here is to highlight that the actuation mechanism of the dual-responsive films is not based on the non-Faradaic reaction, by the comparison between the actuating hypothesis illustration (non-Faradaic) and actual actuating phenomenon (Faradaic).

However, in order to avoid the misunderstanding of these two figures, we supplemented some highlighted labels and subtitles in these two figures to illustrate our ideas more clearly.

Q2. *The mechanism of the pseudocapacitive actuation is novel, can the authors provide more quantification discussion to predict the bending angles of the film under different voltages or nanowires directions. So that this model would be more general and guide other polymer actuators design.*

A: Thanks for the reviewer's suggestion.

(1) Indeed, as the reviewer mentioned, the bending angles are related to the applied voltages. As shown in Fig. 5b and Supplementary Fig. 14d, with the decrease of potentials from -0.2 to -0.9 V, the maximum bending angle and curvature at each potential increased to 32° for -0.2 V, and 215° for -0.9 V. This is mainly related to more Li^+ intercalation into the $\text{W}_{18}\text{O}_{49}$ NWs and further

contraction of the lattice spacing with decrease of the applied potentials. This is also consistent with the result in the *operando* X-ray synchrotron diffraction of $W_{18}O_{49}$ NWs that the lattice spacing will contract obviously even under a small potential (ca. -0.2 V vs. Ag/AgCl) and reach the maximum contraction at ca. -1.05 V vs. Ag/AgCl. The related description has been already provided in pages 18~19 of the revised manuscript.

(2) For the nanowire direction, although there already existed some methods which were used to align the nanowires, it is still very challenging for us to realize such precise alignment control in a short time due to the limited suitable approaches. For instance, the “coffee ring” effect was used to align the nanowires, but it is still hard to control the thickness of the active layer and transfer the as-prepared thin film onto desired substrates. The facile shear stress or capillary force can also be used to improve the orientation degree, but it is also very challenging to achieve precise control of orientation angle in a long range, and thin-film thickness. These two parameters are the key factors to realize the obvious actuation phenomenon and studying the influence of nanowires direction on the actuating performance. So far, Langmuir–Blodgett (LB) technique is one attractive method and is possible to be used to control the alignment precisely, but unfortunately, the specific equipment is needed to obtain a high-quality orientation alignment.

Nonetheless, we think nanowire orientation is a very important parameter to affect the actuating performance and worthy to studying for the next-step research. According to the XRD and HRTEM characterization results, the synthesized $W_{18}O_{49}$ NWs mainly grow along [010] direction. During the Li-ion intercalation process, the crystal lattice contraction also occurs along the direction of [010], which mainly contributes to the macroscopic actuation. Therefore, it is reasonable to hypothesize that orientation of the $W_{18}O_{49}$ NWs can augment the strain (i.e., contraction) in the direction along the crystallographic *b*-axis (perpendicular to the layers in $W_{18}O_{49}$) and highly improve the bending angles. Moreover, we can reduce and even eliminate the strains from other directions and thus significantly improve the controllability of the bending direction of dual-responsive films. Therefore, for our next-step work, we will try to align the arrangement of $W_{18}O_{49}$ NWs and study the influence of orientation direction on the electrochemical actuation performance.

Meanwhile, we describe some future prospects (highlighted in blue) in the page 20 of revised manuscript to highlight the importance of nanowire orientation and provide some ideas for future actuator designs. The related description was provided as following.

“Although novel phenomenon and systematical analysis have been demonstrated in current work, this pseudocapacitive dual-responsive behavior still need further comprehensive study. For instance, similar as the lattice change in this work, the lattice spacing of some other pseudocapacitive materials will also contract/expand during the ion intercalation/de-intercalation. Therefore, this reversible pseudocapacitive actuation can endow these pseudocapacitive materials with stable actuation performance *via* the similar lattice structure change and can spur their development in actuating application fields. In addition, according to the XRD and HRTEM characterizations, synthesized $W_{18}O_{49}$ NWs mainly grow along [010] direction. During the Li^+ intercalation, the crystal lattice contraction also occurs along the direction of [010], which mainly contributes to the macroscopic actuation. Therefore, it is reasonable to hypothesize that orientation of the $W_{18}O_{49}$ NWs can augment the strain (i.e., contraction) in the direction along the crystallographic *b*-axis (perpendicular to the layers in $W_{18}O_{49}$) and highly improve the bending angles. Moreover, we can reduce and even eliminate the strains from other directions, and thus

significantly improve the controllability of the bending directions of dual-responsive films. At last, a lot of conducting polymer actuators also exhibit promising actuating performance based on the similar electrochemical mechanism. According to the better elasticity modulus and mechanical flexibility, hybridizing conducting polymer with inorganic actuating materials could be a promising approach to increasing the mechanical flexibility and actuation capability.”

Q3. To further confirm the mechanism, the film should bend different angles vs different film shape(triangle, round shape)? The authors can further test this.

A: We appreciate the reviewer’s suggestion. To further confirm the mechanism and demonstrate the tunability of actuation based on the current actuator design, we measured the deformation responses of dual-responsive films with three shapes (triangle, rectangular and the circular shapes) according to reviewer’s suggestion. As shown in the Figure R1, when the voltage was applied, the bending deformation occurred at each corner of the triangle soaked in the electrolyte. For rectangular films, they will bend from both left and right sides (short edges), which is caused by the simultaneous contraction of the four corners. This bending deformation is similar with above demonstration for rectangular dual-responsive films which also bend from the short edge at the bottom. To demonstrate the bending actuation from more directions, we prepared a circular dual-responsive film and cut it into three parts along the red dashed lines (as shown in bottom of Figure R1). As expected, once the voltage was applied, the three cut parts bent to the center of circular film.

We also added the related figures into Supplementary Figure 15 and analysis in the page 19 of revised manuscript and highlighted them in blue.

Figure R1. Deformation responses of the dual-responsive films with different shapes. (The red dashed lines represent the active areas for deformation, i.e., the areas of dual-responsive films soaked in the electrolyte; The yellow dashed line represents the Pt counter electrode which is behind the dual-responsive film)

REVIEWERS' COMMENTS:

Reviewer #1 (Remarks to the Author):

The authors have answered all the questions perfectly. It can now be accepted.